# You Never Cluster Alone

Yuming Shen[*1], Ziyi Shen[2], Menghan Wang[3], Jie Qin[4], Philip H.S. Torr[1], and Ling Shao[5]

[1]University of Oxford   [2]University College London   [3]eBay
[4]Nanjing University of Aeronautics and Astronautics   [5]Inception Institute of Artificial Intelligence
ymcidence@gmail.com

## Abstract

Recent advances in self-supervised learning with instance-level contrastive objectives facilitate unsupervised clustering. However, a standalone datum is not perceiving the context of the holistic cluster, and may undergo sub-optimal assignment. In this paper, we extend the mainstream contrastive learning paradigm to a cluster-level scheme, where all the data subjected to the same cluster contribute to a unified representation that encodes the context of each data group. Contrastive learning with this representation then rewards the assignment of each datum. To implement this vision, we propose twin-contrast clustering (TCC). We define a set of categorical variables as clustering assignment confidence, which links the instance-level learning track with the cluster-level one. On one hand, with the corresponding assignment variables being the weight, a weighted aggregation along the data points implements the set representation of a cluster. We further propose heuristic cluster augmentation equivalents to enable cluster-level contrastive learning. On the other hand, we derive the evidence lower-bound of the instance-level contrastive objective with the assignments. By reparametrizing the assignment variables, TCC is trained end-to-end, requiring no alternating steps. Extensive experiments show that TCC outperforms the state-of-the-art on challenging benchmarks.

## 1   Introduction

Ancestored by various similarity-based [59] and feature-based [4, 56] approaches, unsupervised deep clustering jointly optimizes data representations and cluster assignments [81]. A recent fashion in this domain takes inspiration from contrastive learning in computer vision [11, 12, 24], leveraging the effectiveness and simplicity of discriminative feature learning. This strategy is experimentally reasonable, as previous research has found that the learnt representations reveal data semantics and locality [34, 80]. Even a simple migration of contrastive learning significantly improves clustering performance, of which examples include a two-stage clustering pipeline [73] with contrastive pre-training and $k$-means [56] and a composition of an InfoNCE loss [62] and a clustering one [81] in [89]. Compared with the deep generative counterparts [16, 36, 54, 86], contrastive clustering is free from decoding and computationally practical, with guaranteed feature quality.

However, have we been paying too much attention to the representation expressiveness of a single data point? Intuitively, a standalone data point, regardless of its feature quality, cannot tell us much about how the cluster looks like. Fig. 1 illustrates a simple analogy using the *TwoMoons* dataset. Without any context for the crescents, it is difficult to assign a data point to either of the two clusters based on its own representation, as the point can be inside one moon but still close to the other. Accordingly, observing more data reveals more about the holistic distributions of the clusters, *e.g.*, the shapes of the moons in Fig. 1, and thus heuristically benefits clustering. Although we can implicitly parametrize the context of the clusters by the model itself, *e.g.*, using a Gaussian mixture model (GMM) [4] or encoding this information by deep model parameters, explicitly representing the context yields the most common deep learning practice. This further opens the door for learning cluster-level representations with all corresponding data points. *Namely, you never cluster alone.*

---

[*]A part of this work was done when the author was with eBay.

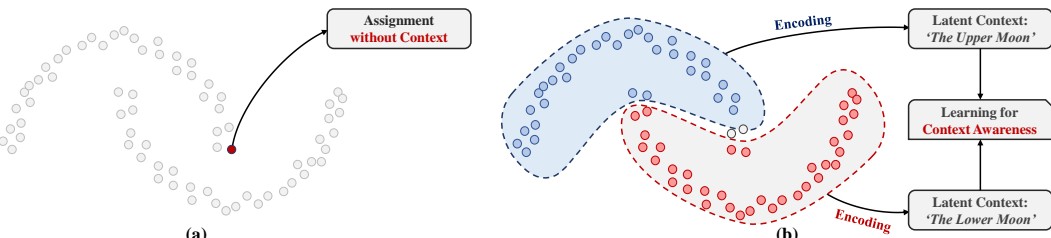

Figure 1: The motivation behind this work. **(a):** When assessing a standalone data point, cluster assignment can be challenging due to the non-linearity of the feature space and the lack of context information for the data distribution. **(b):** Our model learns this context by representing each cluster with latent features.

In this paper, motivated by the thought experiment above, we develop a multi-granularity contrastive learning framework, which includes an instance-level granularity and a cluster-level one. The former learning track conveys the conventional functionalities of contrastive learning, *i.e.*, learning compact data representations and preserving underlying semantics. We further introduce a set of latent variables as cluster assignments, and derive an evidence lower bound (ELBO) of instance-level InfoNCE [62]. As per the cluster-level granularity, we leverage these latent variables as weights to aggregate all the corresponding data representations as the set representation [87] of a cluster. We can then apply contrastive losses to all clusters, thereafter rewarding/updating the cluster assignments. Abbreviated as twin-contrast clustering (TCC), our work delivers the following contributions:

- We develop the novel TCC model, which, for the first time, shapes and leverages a unified representation of the cluster semantics in the context of contrastive clustering.
- We define and implement the cluster-level augmentations in a batch-based training and stochastic labelling procedure, which enables on-the-fly contrastive learning on clusters.
- We achieve significant performance gains against the state-of-the-art methods on five benchmark datasets. Moreover, TCC can be trained from scratch, requiring no pre-trained models or auxiliary knowledge from other domains.

## 2 Preliminaries

### 2.1 Contrastive Learning

Contrastive learning, as the name suggests, aims to distinguish an instance from all others using embeddings, with the dot-product similarity typically being used as the measurement. Let $\mathbf{X} = \{\mathbf{x}_i\}_{i=1}^N$ be an $N$-point dataset, with the $i$-th observation $\mathbf{x}_i \in \mathbb{R}^{d_x}$. An arbitrary transformation $f : \mathbb{R}^{d_x} \to \mathbb{R}^{d_m}$ encodes each data point into a $d_m$-dimensional vector. With the index $i$ being the identifier, the InfoNCE loss [62] discriminates $\mathbf{x}_i$ from the others through a softmax-like likelihood:

$$\log p\left(i | \mathbf{x}_i\right) = \log \frac{\exp\left(\mathbf{v}_i^{\mathsf{T}} f(\mathbf{x}_i)/\tau\right)}{\sum_{j=1}^N \exp\left(\mathbf{v}_j^{\mathsf{T}} f(\mathbf{x}_i)/\tau\right)}. \tag{1}$$

A temperature hyperparameter $\tau$ controls the concentration level [26, 80]. $\mathbf{V} = \{\mathbf{v}_i\}_{i=1}^N$ refers to the vocabulary of the dataset, which is usually based on the data embeddings under different augmentations. As caching the entire $\mathbf{V}$ is not practical for large-scale training, existing works propose surrogates of Eq. (1), *e.g.*, replacing $\mathbf{V}$ with a memory bank [80], queuing it with a momentum network [12, 24], or training with large batches [11].

### 2.2 Deep Set Representations

To learn the representations of sets, we need to consider permutation-invariant transformations. Zaheer *et al.* [87] showed that all permutation-invariant functions $T(\cdot)$ applied to a set $\mathbf{X}$ generally fall into the following form:

$$T(\mathbf{X}) = g\left(\sum_{i=1}^N h(\mathbf{x}_i)\right), \tag{2}$$

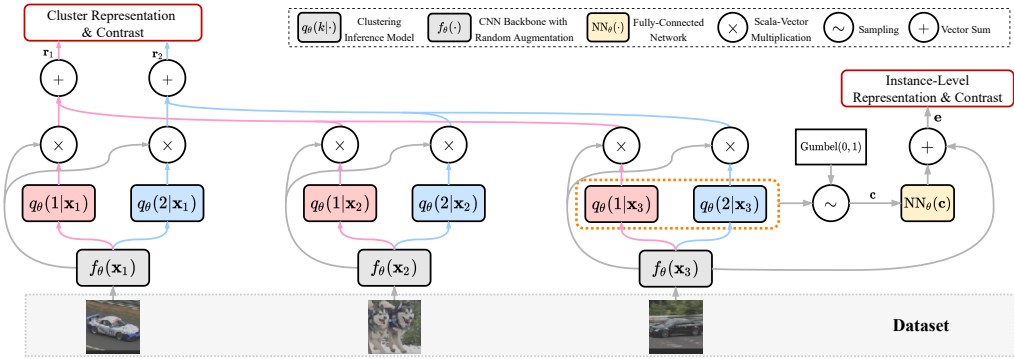

Figure 2: The schematic of TCC. Here, we demonstrate a 2-way clustering example with three images. For simplicity, we only illustrate the instance-level learning module on the last image, while it is applied to all images.

where $g(\cdot)$ and $h(\cdot)$ are arbitrary continuous transformations. Note that the aggregation above can be executed on a weighted basis, which is typically achieved by the attention mechanism between the set-level queries and instance-level keys/values [31, 49]. Our design on each cluster is partially inspired by this, as back-propagation does not support hard instance-level assignment. Next, we define our cluster representation along with the clustering procedure, and describe how it is trained with contrastive learning.

## 3  Twin-Contrast Clustering

We consider a $K$-way clustering problem, with $K$ being the number of clusters. Let $k \in \{1, \cdots, K\}$ denote the entry of a cluster that $\mathbf{x}_i$ may belong to, and the categorical variable $\boldsymbol{\pi}_i = [\pi_i(1), \cdots, \pi_i(K)]$ indicate the cluster assignment probabilities of $\mathbf{x}_i$. Following common practice, we regard image clustering as our target for simplicity. Fig. 2 provides a schematic of TCC. The cluster-level contrast track reflects our motivation from Sec. 1, while the instance-level one learns the semantics of each image. We bridge these two tracks with the inference model $\pi_i(k) = q_\theta(k|\mathbf{x}_i)$ so that both losses reward and update the assignment on each $\mathbf{x}_i$. In particular, $q_\theta(k|\mathbf{x}_i)$ is parametrized by a softmax operation:

$$\pi_i(k) = q_\theta(k|\mathbf{x}_i) = \frac{\exp(\mu_k^\mathsf{T} f_\theta(\mathbf{x}_i))}{\sum_{k'=1}^K \exp(\mu_{k'}^\mathsf{T} f_\theta(\mathbf{x}_i))}, \tag{3}$$

where $f_\theta(\cdot)$ is a convolutional neural network (CNN) [47, 48] built upon random data augmentations, producing $d_m$-dimensional features. We denote $\boldsymbol{\mu}_\theta = \{\mu_k\}_{k=1}^K$ as a set of trainable cluster prototypes, where $\theta$ refers to the collection of all parameters. In Sec. 3.1, we leverage $q_\theta(k|\mathbf{x}_i)$ to aggregate cluster features, and in Sec. 3.2 we derive the ELBO of Eq. (1) with $q_\theta(k|\mathbf{x}_i)$. In the following, we will omit the index $i$ for brevity when $\mathbf{x}$ and $\boldsymbol{\pi}$ clearly correspond to a single data point.

### 3.1  Representing and Augmenting the Context for Cluster-Level Contrast

**Cluster-Level Representation**    We implement Eq. (2) for each cluster using soft aggregation, where $q_\theta(k|\mathbf{x})$ weighs the relevance of the data to the given cluster. Denoted as $\mathbf{r}_k$, the $k$-th cluster representation is computed by:

$$h_\theta(\mathbf{x}; k) = \pi(k) \cdot f_\theta(\mathbf{x}),$$
$$\mathbf{r}_k = T_\theta(\mathbf{X}; k) = \left(\sum_{i=1}^N h_\theta(\mathbf{x}_i; k)\right) \bigg/ \left\|\sum_{i=1}^N h_\theta(\mathbf{x}_i; k)\right\|_2, \tag{4}$$

where $\|\cdot\|_2$ refers to the L2-norm. We adopt L2 normalization here for two main purposes. First, the summation of $\pi(k) = q_\theta(k|\mathbf{x})$ along $\mathbf{x}$ is not self-normalized. Second, and more importantly, as shown in [11, 12, 24], L2-normalized features benefit contrastive learning.

**The Anchored Cluster Semantics**    Intuitively, $\pi(k)$ reflects the degree of relevance of a datum to the $k$-th cluster. With it being the aggregation weight, $\mathbf{r}_k$ represents the information that is related to the corresponding prototype $\mu_k$. In other words, our design treats each $\mu_k$ as the semantic anchor that queries the in-coming batch to form a representation describing a certain latent topic.

**Cluster-Level Augmentation Equivalents**  Contrastive learning is usually employed alongside random data augmentation [11, 24] to obtain positive candidates. Though defining a uniform augmentation scheme for sets is beyond the scope of this paper, the proposed model reflects cluster-level augmentation in its design by the following heuristics:

(a) **Augmentation on elements.** TCC implicitly inherits existing image augmentation techniques (such as cropping, color jittering, random flipping, and grayscale conversion) by implementing them using $f_\theta(\cdot)$.

(b) **Irrelevant minorities.** We consider injecting a small proportion of irrelevant data into a cluster representation, while keeping the main semantics of the cluster unchanged. Eq. (4) turns out to be an equivalent to this. As the softmax product $\pi(k)$ is always positive, those data that are not very related to the given cluster still contribute to the cluster's representation, which compiles the **irrelevant**. Meanwhile, these irrelevant data are not dominating the value of the cluster representation, because the small value of $\pi(k)$ scales the feature magnitude during aggregation, which counts the **minority**.

(c) **Subsetting.** Empirically, a subset of a cluster holds the same semantics as the original. Batch-based training samples data at each step, which naturally creates subsets for each cluster.

We experimentally find the above augmentation equivalents are sufficient for the clustering task. On the other hand, since Eq. (4) is permutation-invariant, reordering the sequence of data does not yield a valid augmentation.

**Cluster-Level Contrastive Learning**  We define a simple contrastive objective that preserves the identity of each cluster against the rest. Having everything in a batch, *e.g.*, a SimCLR-like framework [11], does not allow augmentation (c) to be fully utilized in the loss, since the two augmented counterparts $\mathbf{r}_k$ and $\hat{\mathbf{r}}_k$ from a batch may form part of the same subset of a cluster. Hence, we opt for the MoCo-like solution [24], employing an $L$-sized memory queue $\mathbf{P} = \{\mathbf{p}_l\}_{l=1}^L$ to cache negative samples and a momentum network to produce $\hat{\mathbf{r}}_k = T_{\hat\theta}(\mathbf{X}; k)$. $\mathbf{P}$ stores each cluster representation under different subsets, training with which preserves the temporal semantic consistency [45] of clusters. Our cluster-level objective minimizes the following negative log-likelihood (NLL):

$$\mathcal{L}_1 = \mathbb{E}_k[-\log p_\theta(k|\mathbf{r}_k)] = -\frac{1}{K}\sum_{k=1}^{K} \log \frac{\exp(\hat{\mathbf{r}}_k^\intercal \mathbf{r}_k/\tau)}{\exp(\hat{\mathbf{r}}_k^\intercal \mathbf{r}_k/\tau) + \sum_{l=1}^{L}\exp(\mathbf{p}_l^\intercal \mathbf{r}_k/\tau)\mathbb{1}(l \bmod K \neq k)},$$
(5)

where $\mathbb{1}(\cdot)$ is an indicator function and $\mathrm{mod}$ is the modular operator. Since the cluster number $K$ can be less than the queue size $L$, we exclude the features that represent the same cluster as $k$ in the negative sample collection $\mathbf{P}$ by inserting the indicator function into the loss above.

## 3.2  Instance-Level Contrast with Cluster Assignments

**The ELBO**  We propose to reuse the inference model $q_\theta(k|\mathbf{x})$ discussed above to compute the instance-level contrastive loss, so that the clustering process can benefit from contrastive learning. Let us start from the following ELBO of $\log p_\theta(i|\mathbf{x})$ in Eq. (1):

$$\log p_\theta(i|\mathbf{x}) \geq \mathbb{E}_{q_\theta(k|\mathbf{x})}\left[\log p_\theta(i|\mathbf{x}, k)\right] - \mathrm{KL}\left(q_\theta(k|\mathbf{x})||p_\theta(k|\mathbf{x})\right),$$
(6)

where $\mathrm{KL}(\cdot)$ is the Kullback–Leibler (KL) divergence. We derive this ELBO in Appendix A. The true distribution $p_\theta(k|\mathbf{x})$ is not available under the unsupervised setting. We follow [40, 69] and use a fixed prior instead. In practice, we employ the uniform distribution, *i.e.*, $p_\theta(k|\mathbf{x}) := p_\theta(k) = 1/K$. Then, the KL term above can be reduced to a simple form $-\mathrm{KL}(q_\theta(k|\mathbf{x})||p_\theta(k|\mathbf{x})) = \log K + \mathcal{H}(q_\theta(k|\mathbf{x}))$. Empirically, this encourages an evenly distributed cluster assignment across the dataset.

Regarding the expectation term in Eq. (6), back-propagation through the discrete entry $k$ is not feasible. We resort to the Gumbel softmax trick [32, 57] as relaxation. Specifically, a latent variable $\mathbf{c} \in (0, 1)^K$ is assigned to each $\mathbf{x}$ as a replacement. Each entry $\mathbf{c}(k)$ yields the reparametrization $\mathbf{c}(k) = \mathrm{Softmax}_k((\log \pi(k) + \epsilon(k))/\lambda)$, where $\epsilon \sim \mathrm{Gumbel}(0, 1)$ and $\lambda$ is another temperature hyperparameter. Hence, we obtain the surrogate $\mathbb{E}_{q_\theta(k|\mathbf{x})}\left[\log p_\theta(i|\mathbf{x}, k)\right] \simeq \mathbb{E}_\epsilon\left[\log p_\theta(i|\mathbf{x}, \mathbf{c})\right]$ and the gradients can be estimated with Monte Carlo.

**Instance-Level Contrastive Learning**    In alignment with Eq. (5), $\log p_\theta(i|\mathbf{x}, \mathbf{c})$ learns the representation of $\mathbf{x}$ on a momentum contrast basis [24] by defining the following transformation:

$$\mathbf{e} = (f_\theta(\mathbf{x}) + \mathrm{NN}_\theta(\mathbf{c}))/\|f_\theta(\mathbf{x}) + \mathrm{NN}_\theta(\mathbf{c})\|_2,$$

$$\log p_\theta(i|\mathbf{x}, \mathbf{c}) = \log \frac{\exp(\hat{\mathbf{e}}^\mathsf{T}\mathbf{e}/\tau)}{\exp(\hat{\mathbf{e}}^\mathsf{T}\mathbf{e}/\tau) + \sum_{j=1}^{J} \exp(\mathbf{q}_j^\mathsf{T}\mathbf{e}/\tau)}, \quad (7)$$

where $\mathrm{NN}_\theta(\cdot)$ denotes a single fully connected network. We accordingly use $\hat{\mathbf{e}}$ to indicate the representation of $\mathbf{x}$ processed by a momentum network $f_{\hat{\theta}}(\cdot)$ and $\mathrm{NN}_{\hat{\theta}}(\cdot)$ under different augmentations and Gumbel samplings [32, 57]. A $J$-sized memory queue $\mathbf{Q} = \{\mathbf{q}_j\}_{j=1}^{J}$ is also introduced to cache negative samples, updated by $\hat{\mathbf{e}}$. In this way, we obtain the instance-level loss:

$$\mathcal{L}_2 = \mathbb{E}_i[-\mathbb{E}_{\epsilon_i}[\log p_\theta(i|\mathbf{x}_i, \mathbf{c}_i)] - \mathcal{H}(q_\theta(k|\mathbf{x}_i)) - \log K]. \quad (8)$$

## 3.3   Training and Inference

TCC enables end-to-end training from scratch. Our learning objective is a simple convex combination of Eq. (5) and (8), *i.e.*,

$$\mathcal{L} = \alpha\mathcal{L}_1 + (1-\alpha)\mathcal{L}_2. \quad (9)$$

The hyperparameter $\alpha \in (0, 1)$ controls the contributions of the two contrastive learning tracks. As discussed in Sec. 3.1, $\mathcal{L}$ is computed following a batch-based routine. For each data point $\mathbf{x}$, we obtain only one sample $\mathbf{c}$ from the Gumbel distribution at each step, since this is usually sufficient for long-term training [39, 40, 69]. One may also regard this stochasticity as an alternative to data augmentation. The overall training algorithm is shown in Alg. 1. Here, $\mathbf{\Gamma}(\cdot)$ indicates an arbitrary stochastic gradient descent (SGD) optimizer.

---
**Algorithm 1:** Training Algorithm of TCC

---
**Input:** Dataset $\mathbf{X} = \{\mathbf{x}_i\}_{i=1}^{N}$
**Output:** Network parameters $\theta$.
Initialize $\hat{\theta} = \theta$
**repeat**
  Randomly select a mini-batch from $\mathbf{X}$
  **for** *each $\mathbf{x}_i$ in the batch* **do**
    Randomly augment $\mathbf{x}_i$ twice
    Sample $\mathbf{c}_i$ with Gumbel softmax
  **end**
  $\mathcal{L} \leftarrow$ Eq. (9)
  $\theta \leftarrow \theta - \mathbf{\Gamma}(\nabla_\theta\mathcal{L})$
  Update the queues $\mathbf{P}$ with $\hat{\mathbf{r}}$ and $\mathbf{Q}$ with $\hat{\mathbf{e}}$
  Update $\hat{\theta}$ with momentum moving average
**until** *convergence or reaching max iteration*;

---

All trainable components are subscripted by $\theta$, while those marked with $\hat{\theta}$ are the network momentum counterparts to be updated with momentum moving average. Inference with TCC only requires disabling random data augmentation and then computing $\arg\max_k q_\theta(k|\mathbf{x})$.

**Complexity**    When sampling once for each $\mathbf{x}$ during training, the time complexity for Eq. (9) is $\mathcal{O}(L+J)$, while the memory complexity for the memory bank turns out to be the same. Here we omit the complexity introduced by the CNN backbone and dot-product computation, as it is orthogonal to the design. Compared with the recent mixture-of-expert approach [73], which requires a time and memory complexity of $\mathcal{O}(KJ)$, TCC is trained in a more efficient way.

## 3.4   Relations to Existing Works

MiCE [73] also proposes a lower bound for the instance-level contrastive objective. However, it does not directly *reparametrize* the variational model $q_\theta(k|\mathbf{x})$ for lower-bound computation and inference, but instead employs a $K$-expert solution with EM. This design is less efficient than TCC since each data point needs to be processed by all $K$ experts. Moreover, MiCE [73] does not consider cluster-level discriminability. SCL [28] follows a similar motivation to TCC in cluster-level discriminability, but it implements this with an instance-to-set similarity, while our model learns a unified representation for each cluster. Furthermore, in SCL [28], the clustering inference model is disentangled from the instance-level contrastive objective. In contrast, the inference model $q_\theta(k|\mathbf{x})$ of TCC contributes to instance-level discrimination (Eq. (6)).

We recently find CC [53] comes with a cluster-level contrastive loss as well. It utilizes the in-batch inference results $[\pi_1(k), \cdots, \pi_n(k)]$ to describe the $k$-th cluster. However, this procedure is not literally learning the cluster representation, since it is not permutation free. Re-ordering the batch may shift the semantics of the produced feature. We mitigate this issue with deep sets [87] and the empirical cluster-level augmentations for temporal consistency [45]. A similar problem is witnessed in [66]. In addition, our instance-level discrimination model yields a more general case than the one

of CC [53]. When removing the stochasticity and enforcing $p_\theta(i|\mathbf{x}, \mathbf{c}) \coloneqq p_\theta(i|\mathbf{c})$ in our model, $\mathcal{L}_2$ reduces to the one of [53]. We experimentally show that our design preserves more data semantics, and thus benefits clustering. Being not related to our main contribution, we provide more elaboration on this in Appendix B under the framework of variational information bottleneck [2].

## 4   Related Work

**Deep Clustering**   In addition to the classic approaches [4, 13, 18, 20, 43, 56, 59, 78, 91], the concept of simultaneous feature learning and clustering with deep models can be traced back to [81, 83]. The successors, including [9, 23, 61, 68, 77, 79, 85], have continuously improved the performance since. As a conventional option for unsupervised learning, deep generative models are also widely adopted in clustering [10, 16, 35, 36, 42, 51, 58, 86, 90], usually backboned by VAE [39] and GAN [21]. However, generators are computationally expensive for end-to-end training, and often less effective than the discriminative models [15, 27, 33] in feature learning [11]. Recent research has considered contrastive learning in clustering [28, 53, 73, 74, 89]. We discuss the drawback of them and their relations to TCC in Sec. 3.4.

**Contrastive Learning**   Contrastive learning learns compact image representations in a self-supervised manner [11, 12, 24, 62, 72]. There are various applications for this paradigm [34, 41, 82, 93]. We note that several contrastive learning approaches [7, 52] conceptually involve a clustering procedure. Nevertheless, they are based on a unified pre-training framework to benefit the downstream tasks, instead of delivering a specific clustering model.

**Set Representations**   Our cluster-level representation (Eq. (4)) is a realization of deep sets [17, 87]. Existing research in this area mainly focuses on set-level tasks [19, 31, 37, 76, 84]. It is also notable that, though we leverage cluster-level representation learning, TCC is still an instance-level clustering model, which is different from the set-level clustering models [49, 50, 63].

## 5   Experiments

### 5.1   Settings

We follow the recent works [29, 33] and report the performance of TCC in terms of clustering accuracy (ACC) [81], normalized mutual information (NMI) [70] and adjusted random index (ARI) [30]. For fair comparison with existing works, we do not use any supervised pre-trained models. The experiments are conducted

Table 1: Dataset settings for our experiments.

| Dataset | Images | Clusters $(K)$ | Input Size |
|---|---|---|---|
| CIFAR-10 [44] | 60,000 | 10 | $32 \times 32$ |
| CIFAR-100 [44] | 60,000 | 20 | $32 \times 32$ |
| STL-10 [14] | 13,000 | 10 | $96 \times 96$ |
| ImageNet-10 [9] | 13,000 | 10 | $96 \times 96$ |
| ImageNet-Dog [9] | 19,500 | 15 | $96 \times 96$ |

on five benchmark datasets, including CIFAR-10/100 [44], ImageNet-10/Dog [9] and STL-10 [14]. Note that ImageNet-10/Dog [9] is a subset of the original ImageNet dataset [67]. Since most existing works have pre-defined cluster numbers, we adopt this practice and follow their training/test protocols [29, 33, 61, 73]. Tab. 1 depicts the details of the settings.

### 5.2   Implementation Details

TCC is implemented with the deep learning toolbox TensorFlow [1]. We choose the MoCo-style random image augmentations [24] for fair comparison with the recent works [28, 73]. We further link our choice of augmentations with the cluster representation temporal consistency in Sec. 3.1. Specifically, each image is successively processed by random cropping, gray-scaling, color jittering, and horizontal flipping, followed by mean-std standardization. We refer to [24, 73] for more details. We employ ResNet-34 [25] as the default CNN backbone $f_\theta(\cdot)$, which is also identical to [28, 73]. Appendix C gives a full illustration of the CNN structure. The image size $d_x$ and cluster number $K$ are fixed for each dataset, as shown in Tab. 1. The feature dimensionality produced by CNN is $d_m = 128$. Following common practice [12, 24, 11], we fix the contrastive temperature $\tau = 1$, while using a slightly lower $\lambda = 0.8$ for the Gumbel softmax trick [32, 57] to encourage concrete assignments. We implement a fixed-length instance-level memory bank $\mathbf{Q}$ with a size of $J = 12,800$ to match up with the smallest dataset in our experiments. The size of the cluster-level memory bank $\mathbf{P}$ is set to $L = 100 \times K$, varying from each dataset. We have $\alpha = 0.5$ so that $\mathcal{L}_1$ and $\mathcal{L}_2$ provide equal contributions to training. The choice of batch size is of importance to TCC in computing

Table 2: Unsupervised clustering performance comparison with existing methods (in percentage %). We provide additional results on Tiny ImageNet [46] and comparison with more contrastive baselines such as SwAV [7] in Appendix D.

| Method | CIFAR-10 | | | CIFAR-100 | | | STL-10 | | | ImageNet-10 | | | ImageNet-Dog | | |
|---|---|---|---|---|---|---|---|---|---|---|---|---|---|---|---|
| | NMI | ACC | ARI | NMI | ACC | ARI | NMI | ACC | ARI | NMI | ACC | ARI | NMI | ACC | ARI |
| AC [22] | 10.5 | 22.8 | 6.5 | 9.8 | 13.8 | 3.4 | 23.9 | 33.2 | 14.0 | 13.8 | 24.2 | 6.7 | 3.7 | 13.9 | 2.1 |
| NMF [5] | 8.1 | 19.0 | 3.4 | 7.9 | 11.8 | 2.6 | 9.6 | 18.0 | 4.6 | 13.2 | 23.0 | 6.5 | 4.4 | 11.8 | 1.6 |
| AE [3] | 23.9 | 31.4 | 16.9 | 10.0 | 16.5 | 4.8 | 25.0 | 30.3 | 16.1 | 21.0 | 31.7 | 15.2 | 10.4 | 18.5 | 7.3 |
| DAE [75] | 25.1 | 29.7 | 16.3 | 11.1 | 15.1 | 4.6 | 22.4 | 30.2 | 15.2 | 20.6 | 30.4 | 13.8 | 10.4 | 19.0 | 7.8 |
| DCGAN [65] | 26.5 | 31.5 | 17.6 | 12.0 | 15.1 | 4.5 | 21.0 | 29.8 | 13.9 | 22.5 | 34.6 | 15.7 | 12.1 | 17.4 | 7.8 |
| DeCNN [88] | 24.0 | 28.2 | 17.4 | 9.2 | 13.3 | 3.8 | 22.7 | 29.9 | 16.2 | 18.6 | 31.3 | 14.2 | 9.8 | 17.5 | 7.3 |
| VAE [39] | 24.5 | 29.1 | 16.7 | 10.8 | 15.2 | 4.0 | 20.0 | 28.2 | 14.6 | 19.3 | 33.4 | 16.8 | 10.7 | 17.9 | 7.9 |
| JULE [85] | 19.2 | 27.2 | 13.8 | 10.3 | 13.7 | 3.3 | 18.2 | 27.7 | 16.4 | 17.5 | 30.0 | 13.8 | 5.4 | 13.8 | 2.8 |
| DEC [81] | 25.7 | 30.1 | 16.1 | 13.6 | 18.5 | 5.0 | 27.6 | 35.9 | 18.6 | 28.2 | 38.1 | 20.3 | 12.2 | 19.5 | 7.9 |
| DAC [9] | 39.6 | 52.2 | 30.6 | 18.5 | 23.8 | 8.8 | 36.6 | 47.0 | 25.7 | 39.4 | 52.7 | 30.2 | 21.9 | 27.5 | 11.1 |
| ADC [23] | - | 32.5 | - | - | 16.0 | - | - | 53.0 | - | - | - | - | - | - | - |
| DDC [8] | 42.4 | 52.4 | 32.9 | - | - | - | 37.1 | 48.9 | 26.7 | 43.3 | 57.7 | 34.5 | - | - | - |
| DCCM [79] | 49.6 | 62.3 | 40.8 | 28.5 | 32.7 | 17.3 | 37.6 | 48.2 | 26.2 | 60.8 | 71.0 | 55.5 | 32.1 | 38.3 | 18.2 |
| IIC [33] | 51.3 | 61.7 | 41.1 | - | 25.7 | - | 43.1 | 49.9 | 29.5 | - | - | - | - | - | - |
| MMDC [68] | 57.2 | 70.0 | - | 25.9 | 31.2 | - | 49.8 | 61.1 | - | 71.9 | 81.1 | - | 27.4 | 11.9 | - |
| PICA [29] | 56.1 | 64.5 | 46.7 | 29.6 | 32.2 | 15.9 | - | - | - | 78.2 | 85.0 | 73.3 | 33.6 | 32.4 | 17.9 |
| DCCS [92] | 56.9 | 65.6 | 46.9 | - | - | - | 37.6 | 48.2 | 26.2 | 60.8 | 71.0 | 55.5 | - | - | - |
| DHOG [15] | 58.5 | 66.6 | 49.2 | 25.8 | 26.1 | 11.8 | 41.3 | 48.3 | 27.2 | - | - | - | - | - | - |
| GATCluster [61] | 47.5 | 61.0 | 40.2 | 21.5 | 28.1 | 11.6 | 44.6 | 58.3 | 36.3 | 59.4 | 73.9 | 55.2 | 28.1 | 32.2 | 16.3 |
| IDFD [71] | 71.4 | 81.5 | 66.3 | 42.6 | 42.5 | 26.4 | 64.3 | 75.6 | 57.5 | **89.8** | **95.4** | **90.1** | 54.6 | 59.1 | 41.3 |
| CC [53] | 70.5 | 79.0 | 63.7 | 43.1 | 42.9 | 26.6 | **76.4** | **85.0** | **72.6** | 85.9 | 89.3 | 82.2 | 44.5 | 42.9 | 27.4 |
| MoCo baseline [73] | 66.9 | 77.6 | 60.8 | 39.0 | 39.7 | 24.2 | 61.5 | 72.8 | 52.4 | - | - | - | 34.7 | 33.8 | 19.7 |
| MiCE [73] | 73.7 | 83.5 | 69.8 | 43.6 | 44.0 | 28.0 | 63.5 | 75.2 | 57.5 | - | - | - | 42.3 | 43.9 | 28.6 |
| TCC | **79.0** | **90.6** | **73.3** | **47.9** | **49.1** | **31.2** | 73.2 | 81.4 | 68.9 | 84.8 | 89.7 | 82.5 | **55.4** | **59.5** | **41.7** |

the cluster-level representations. We set it to $32 \times K$ by default to ensure that sufficient images can be assigned to a cluster at each step. Training TCC only requires SGD *w.r.t.* $\theta$ and momentum update *w.r.t.* $\hat{\theta}$. We employ the Adam optimizer [38] with a default learning rate of $3 \times 10^{-3}$, without learning rate scheduling. The momentum network is updated by $\hat{\theta} \leftarrow 0.999\hat{\theta} + 0.001\theta$, where all modules subscripted by $\hat{\theta}$ are involved in this procedure. We train TCC for at least $1,000$ epochs on a single NVIDIA V100 GPU.

## 5.3 Comparison with the State-of-the-Art

**Baselines** Both deep clustering and traditional models are compared, including a MoCO-based two-stage baseline introduced by [73]. Similar to the recent works [29, 33, 73, 79], we pick deep models that enable training from scratch and do not require supervised pre-training parameters, for fair and reasonable comparison. For this reason, baselines such as VaDE [36] and SPICE [60] are not included here. We also exclude the clustering refinement approaches [64] from our comparison as they are orthogonal to our design.

**Results** The clustering performance (in percentage %) is shown in Tab. 2. For those baselines that are not designed for clustering [3, 21, 39], we report the results with $k$-means on the produced features. TCC outperforms existing works on most benchmarks. In particular, on CIFAR-10 [44], TCC outperforms the state-of-the-art methods by large margins, *e.g.*, 7% higher in ACC than the second best one (*i.e.*, MiCE [73] with even stronger augmentations [11]). As a closely-related work, MiCE [73] only considers instance-level representation learning. The performance gain of TCC over MiCE endorses our motivation to introduce cluster-level representations. We also observe that TCC underperforms [53] on STL-10 [14] due to the exceptionally high performance of it on this dataset. However, TCC is still the runner-up on this dataset by a significant margin, and is superior to [53] on the other four datasets. We

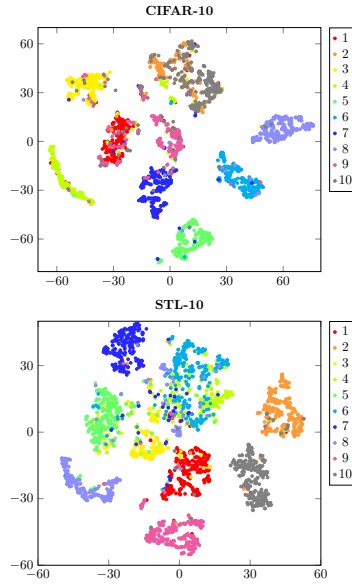

Figure 3: t-SNE visualization on CIFAR-10 [44] and STL-10 [14].

argue that the performance on larger datasets is of more importance when comparing contrastive deep clustering methods, as contrastive learning is originally designed for large-scale tasks. Fig. 3 illustrates the t-SNE [55] scattering results of TCC on CIFAR-10 [44] and STL-10 [14].

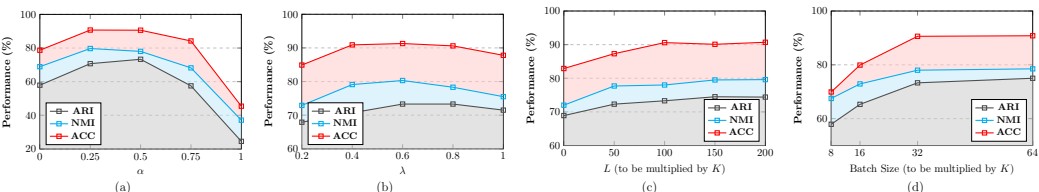

Figure 4: Hyperparameter analysis results on CIFAR-10 [44].

## 5.4 Ablation Study

We conduct an ablation study to validate our motivation and design, with the following baselines.

(i) **Without $\mathcal{L}_1$.** As a key component of TCC, the cluster-level contrastive learning objective $\mathcal{L}_1$ reflects our main motivation. We first assess the model performance when removing this loss, which reduces TCC to a simple instance-level contrastive clustering model.

(ii) **Without $\mathcal{L}_2$.** We can also remove $\mathcal{L}_2$ to see if $\mathcal{L}_1$ alone still yields a valid baseline.

(iii) **Multiple Sampling.** As described in Sec. 3.3, we only consider a single sample $\mathbf{c}$ each time to compute the lower bound of the instance-level loss (Eq. (8)). We also consider applying the Gumbel softmax trick multiple times for each image. In particular, we sample 10 groups of latent variables each time to compute the expectation term $\mathbb{E}_\epsilon \left[ \log p_\theta(i|\mathbf{x}, \mathbf{c}) \right]$ of $\mathcal{L}_2$. On each batch, we enqueue the mean of $\hat{\mathbf{e}}$ *w.r.t.* all 10 sampled $\mathbf{c}$ for each image.

(iv) **Without P.** Since we usually have a small cluster number $K$, computing the cluster-level InfoNCE loss does not necessarily require a memory bank to cache the negative sample surrogates. In this baseline, we remove the cluster-level memory bank $\mathbf{P}$, and use the remaining $K - 1$ cluster representations as negative samples when computing Eq. (5).

(v) **Without Augmentation (a) for $\mathcal{L}_1$.** We validate our cluster-level augmentation strategies by removing image augmentations when computing Eq. (8). Note that this baseline does not influence the instance-level objective by rendering both augmented and original images to $f_\theta(\cdot)$.

(vi) **Without Augmentation (b) for $\mathcal{L}_1$.** This baseline requires hard assignments at each step so that cluster-level aggregation only involves images that are assigned to the corresponding clusters. This modification does not affect $\mathcal{L}_2$.

(vii) **Without Augmentation (c) for $\mathcal{L}_1$.** The final baseline changes the training pipeline. Since we do not subset any clusters here, cluster representation aggregation (Eq. (4)) runs on the whole training set after each epoch. We apply alternating training procedure as follows. First, $\mathcal{L}_2$ optimizes the model for a full epoch. Then, we descend $\mathcal{L}_1$ with aggregated cluster features and repeat.

**Baseline Comparison Results** We show the ablation study results in Tab. 3. Without $\mathcal{L}_1$, TCC performs similarly to the two-stage baseline with MoCo [24] and $k$-means [56] reported in [73], which is slightly lower than MiCE [73]. Interestingly, $\mathcal{L}_1$ does not provide instance-level discriminative information. Although it does still serve as a valid baseline, it does not perform very well (Baseline (ii)). Specifically, we

Table 3: Ablation study results (in percentage %).

| | Baseline | CIFAR-10 | | |
|---|---|---|---|---|
| | | NMI | ACC | ARI |
| (i) | Without $\mathcal{L}_1$ | 68.9 | 78.7 | 57.9 |
| (ii) | Without $\mathcal{L}_2$ | 37.1 | 45.4 | 24.5 |
| (iii) | Multiple Sampling | 78.5 | 90.1 | 74.2 |
| (iv) | Without $\mathbf{P}$ | 72.0 | 82.9 | 68.8 |
| (v) | Without Augmentation (a) for $\mathcal{L}_1$ | 73.5 | 85.3 | 69.1 |
| (vi) | Without Augmentation (b) for $\mathcal{L}_1$ | 68.5 | 79.2 | 60.6 |
| (vii) | Without Augmentation (c) for $\mathcal{L}_1$ | 69.4 | 80.0 | 62.7 |
| | TCC Full Model | 79.0 | 90.6 | 73.3 |

experience strong degeneracy [6, 33] with this baseline, but it still produces better results than the traditional models. We also observe that having multiple samples with the Gumbel softmax trick [32, 57] for gradient estimation does not make much difference from a single-sample solution. Baseline (iv) also underperforms the original model. As discussed in previous sections, having a memory bank for cluster representations provides a way to acquire more negative samples for contrastive learning, considering the fact that the cluster number is usually limited.

**Hyperparameters** We evaluate the hyperparameters most essential to our design, including the loss weight $\alpha$, the temperature of the Gumbel softmax $\lambda$, the cluster-level memory queue length $L$, and the batch size. The InfoNCE temperature $\tau$ and the instance-level memory queue length $J$ are not included here since they are not relevant to our key motivation and have been employed and evaluated in the recent works [24, 28, 73]. The corresponding results are plotted in Fig. 4. Though $\mathcal{L}_1$ plays an

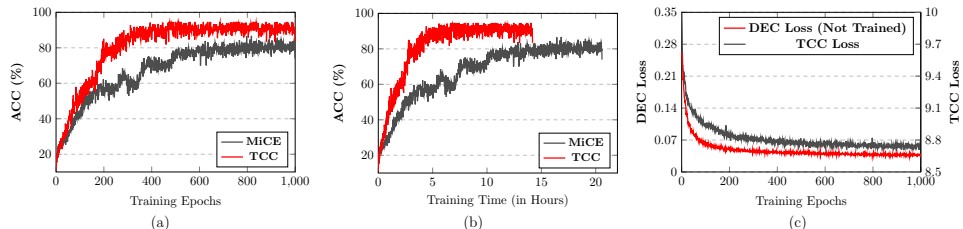

Figure 5: **(a) and (b):** On-batch ACC comparison between TCC and MiCE [73] *w.r.t.* training epochs and training time respectively on CIFAR-10 [44]. **(c):** The values DEC loss [81] during training. Note that TCC is not trained with DEC, and we just record the figures for illustration.

essential role in the proposed model, having large values of $\alpha$ does not improve the performance, as the key instance-level semantics are yet learnt by $\mathcal{L}_2$. Only a reasonable proportion of $\mathcal{L}_1$, *e.g.*, $\alpha = 0.25$ or $0.5$, in the overall learning objective improves the performance of our model. Further, we find that TCC is not very sensitive to the Gumbel softmax temperature $\lambda$, while a moderate hardness of the softmax produces the best results. Empirically, a large batch size benefits TCC, since more data can be involved in the subset of each cluster. Hence the aggregated features on each batch can be more representative. Fig. 4 (d) endorses this intuition. However, training with extremely large batch sizes may lead to out-of-memory problems with large images. To enable training on a single device, we opt to have a fixed batch size of $32 \times K$ in all experiments.

## 5.5 More Results

**Training Time** We compare the training epochs (Fig. 5 (a)) and training time (Fig. 5 (a)) of TCC and the re-implemented version of MiCE [73] with the same optimizer setting. As discussed in Sec. 3.3, MiCE obtains a higher time complexity during training than TCC. This is reflected in Fig. 5 (b), though not linearly proportional. In addition, TCC requires less training steps than MiCE to reach the best-performing results.

**Conventional Clustering Losses** During training, we also cache and observe the DEC loss [81], but we are **not** optimizing the model with it. In Fig. 5 (c), we show that by minimizing $\mathcal{L}$ (Eq. (9)), the traditional DEC loss [81] also decreases. This implicitly endorse our design.

**Assumptions in Design** One merit of constrastive learning is that one does not need to assume any empirical prior distribution to the feature space, which benefits

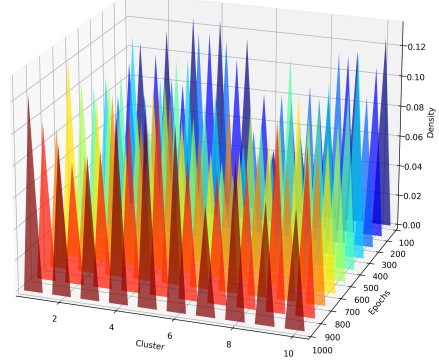

Figure 6: Histograms of cluster assignments during training on CIFAR-10 [44].

TCC when learning the cluster-level representations. The only assumption we employ is that the true posterior $p_\theta(k|\mathbf{x})$ should be uniform to simplify the computation of the KL-divergence in Eq. (6). As previously discussed, this conventional relaxation [69] is intuitively valid since we generally expect evenly assigned clusters. It is illustrated in Fig. 6 that TCC achieves this during training by minimizing $\mathrm{KL}\left(q_\theta(k|\mathbf{x})||p_\theta(k|\mathbf{x})\right) = -\log K - \mathcal{H}(q_\theta(k|\mathbf{x}))$.

## 6 Conclusion

Inspired by the recent success of self-supervised learning, this paper proposes a multi-granularity contrastive clustering framework to exploit the holistic context of a cluster in an unsupervised manner. The proposed TCC simultaneously learns instance- and cluster-level representations by leveraging cluster assignment variables. Cluster-level augmentation equivalents are derived to enable on-the-fly contrastive learning on clusters. Moreover, by reparametrizing the assignment variables, TCC can be trained end-to-end without auxiliary steps. Extensive experiments validate the superiority of TCC, which consistently outperforms competitors on the five benchmarks often by large margins, echoing our major motivation, *i.e.*, we are not clustering alone.

# Acknowledgments

This work is supported by the UKRI grant: Turing AI Fellowship EP/W002981/1 and EPSRC/MURI grant: EP/N019474/1. We also acknowledge the philanthropic support of the donors to the University of Oxford's COVID-19 Research Response Fund: BRD00230. We would like to thank the Royal Academy of Engineering and FiveAI.

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
