# You Never Cluster Alone

Supplementary Material

## Summary of Revision

We introduce several additional unsupervised clustering articles, including deep consensus clustering, IDFD and SCAN [13, 16, 18].

We slightly improve the statement regarding the cluster-wise augmentations according to the reviewers' responses.

We provide additional results on Tiny ImageNet [8] and comparison with more contrastive baselines such as SwAV [2] in Appendix D.

## A    The Instance-Level ELBO

Being not the main contribution of this paper, we are detailing our instance-level contrastive ELBO in the appendix to keep the main content concise. Here we reuse the inference model $q_\theta(k|\mathbf{x})$ as the pivot to bridge the cluster-level objective with the instance-level one. Similar to many VAE-like objectives [6], our ELBO can be derived with the Jensen's inequality $\log \mathbb{E}[\cdot] \geq \mathbb{E}[\log(\cdot)]$ as:

$$
\begin{aligned}
\log p_\theta(i|\mathbf{x}) &= \log \sum_{k=1}^{K} p_\theta(i, k|\mathbf{x}) \\
&= \log \sum_{k=1}^{K} p_\theta(i|\mathbf{x}, k) p_\theta(k|\mathbf{x}) \frac{q_\theta(k|\mathbf{x})}{q_\theta(k|\mathbf{x})} \\
&= \log \mathbb{E}_{q_\theta(k|\mathbf{x})} \left[ p_\theta(i|\mathbf{x}, k) \frac{p_\theta(k|\mathbf{x})}{q_\theta(k|\mathbf{x})} \right] \\
&\geq \mathbb{E}_{q_\theta(k|\mathbf{x})} \left[ \log p_\theta(i|\mathbf{x}, k) \right] + \mathbb{E}_{q_\theta(k|x)} \left[ \log \frac{p_\theta(k|\mathbf{x})}{q_\theta(k|\mathbf{x})} \right] \\
&= \mathbb{E}_{q_\theta(k|\mathbf{x})} \left[ \log p_\theta(i|\mathbf{x}, k) \right] - \mathrm{KL}(q_\theta(k|\mathbf{x}) || p_\theta(k|\mathbf{x})) \\
&= \text{Eq. (6) in the main body.}
\end{aligned}
\tag{1}
$$

As discussed in the main body, the true distribution is $p_\theta(k|\mathbf{x})$ is unavailable in unsupervised learning. We use a uniform distribution as surrogate $p_\theta(k|\mathbf{x}) = p(k) = 1/K$. We also apply the Gumbel softmax trick [4, 11] as a relaxation to replace $\mathbb{E}_{q_\theta(k|\mathbf{x})}[\cdot]$. Thus, the gradients can be estimated by the reparametrization trick [6]. We write the full form of the final ELBO after relaxation as:

$$
\mathbb{E}_\epsilon \left[ \log p_\theta(i|\mathbf{x}, \mathbf{c}) \right] - \mathrm{KL}(q_\theta(k|\mathbf{x}) || p(k)).
\tag{2}
$$

In our experiments, we sample a single Gumbel random variable $\epsilon$ for each image for TCC so that we can use a single-track memory bank for contrastive learning. This can be regarded as a stochastic layer added to the model for additional random augmentation. A similar spirit has been witnessed in [7] where a random noise model applies to enhance the robustness of the learnt representation. Sec. 5.4 shows that a one-sample solution obtains on-par performance as the multiple sampling baseline.

# B  Detailed Relations to Existing Models

We elaborate on the relations and differences between TCC and the most recent works [10, 17]. It is notable that we have discussed them in Sec. 3.4, and here provides more technical details. In addition, by the time of writing, [10] is yet a pre-print version.

## B.1  MiCE [17]

We notice that [17] leverages a CVAE-like [15] lower-bound

$$\mathbb{E}_{q_\theta(k|\mathbf{x},i)}\left[\log p_\theta(i|\mathbf{x},k)\right] - \mathrm{KL}(q_\theta(k|\mathbf{x},i)||p_\theta(k|\mathbf{x})) \tag{3}$$

for instance-level contrast, and analytically computes the expectation term. Inference with MiCE yields a $K$-fold inference solution to obtain:

$$q_\theta(k|\mathbf{x},i) = \frac{p_\theta(k|\mathbf{x})p_\theta(i|\mathbf{x},k)}{\sum_{k'=1}^{K} p_\theta(k'|\mathbf{x})p_\theta(i|\mathbf{x},k')} \tag{4}$$

TCC derives a simpler version of instance-level contrast, and directly parametrizes the inference model $q_\theta(k|\mathbf{x})$ with neural networks. Note that the inference model in Eq. (3) directly describes a simple fully-connected layer with softmax activation. We show the complexity in Sec. 3.3.

On the other hand, MiCE does not consider cluster-level context learning, which counts the main contribution of this paper.

## B.2  CC [10]

CC [10] introduces similar concepts as this paper in terms of **(1)** instance-level contrast with assignments and **(2)** literally 'cluster-level' contrast.

### B.2.1  Instance-Level Objectives as VIB [1]

We firstly show the relation between TCC (proposed) and CC [10] in the instance-level objective with the help of variational information bottleneck (VIB) [1].

If we remove the dependency between the likelihood $p_\theta(i|\mathbf{x},\mathbf{c})$ and $\mathbf{x}$, in Eq. (2) to have $p_\theta(i|\mathbf{x},\mathbf{c}) := p_\theta(i|\mathbf{c})$, the expectation of our instance-level ELBO becomes:

$$\mathbb{E}_{\mathbf{x}}\left[\mathbb{E}_\epsilon\left[\log p_\theta(i|\mathbf{c})\right] - \mathrm{KL}(q_\theta(k|\mathbf{x})||p(k))\right] \leq \mathcal{I}(I,C) - \mathcal{I}(C,X), \tag{5}$$

which is a special case of the ELBO of the information bottleneck $\mathcal{I}(I,C) - \beta\mathcal{I}(C,X)$ [1], with $\beta = 1$. CC [10] considers an even simpler deterministic version of this by removing the stochasticity. To this extent, the instance-level objective of CC can be regarded as a simplified version of our model.

The main difference here is whether we make the contrastive loss dependent on the image $\mathbf{x}$. We argue this dependency is of importance to inject discriminative information to identify each image. The category variables alone are not revealing the instance-level identities. This vision is also endorsed by MiCE [17]. In our experiments, we have shown that even without the cluster-level loss, TCC still obtains similar performance as CC [10] on CIFAR-10 (Sec. 5.4). Note that CC [10] comes with stronger random augmentations and is trained with larger image sizes. This result implicitly legitimates our instance-level design.

### B.2.2  Cluster-Level Objectives and Representations

Literally, CC [10] involves a cluster-level contrastive learning process. Hence, we compare the cluster representations as follows:

$$\text{CC [10]: } \mathbf{r}_k = [\pi_1(k), \cdots, \pi_B(k)]$$

$$\text{TCC (Ours): } \mathbf{r}_k = \text{L2Normalize}\left(\sum_{i=1}^{B} \pi_i(k)f_\theta(\mathbf{x}_i)\right). \tag{6}$$

We use $B$ to indicate the batch size. For CC [10], the degrees of relevance of the data in the batch to a cluster shape the corresponding representation. Note that this solution is not reflecting the latent

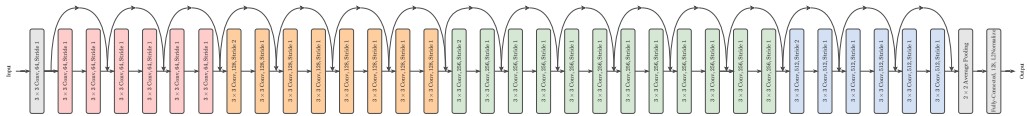

Figure 1: CNN backbone structure.

semantics of a cluster, as reordering a batch would result in absolutely different vectors. The proposed TCC aggregates the features $f_\theta(\mathbf{x})$ according to the relevance $\pi(k)$ queried by the corresponding prototypes $\{\mu_k\}$. Reordering a batch does not change the content of the resulted $\mathbf{r}_k$. In Sec. 3.1, we have discussed that our aggregation is **semantically anchored** by the prototypes $\{\mu_k\}$. Though the actual values throughout different batches are slightly different, our cluster representations always reflect the same latent topics that are defined by $\{\mu_k\}$.

Our design is similar to *pooling by multihead attention* (PMA) in set transformer [9], which is also related to deep set representations [20]. The only difference here is that we are not normalizing the 'attention weights' $\pi_i(k)$ along the axis of $i$ with softmax, as we are not expecting any datum to dominate the final cluster representation.

## C   CNN Backbone

Fig. 1 provides the details of the CNN backbone used in our experiments. All convolutional layers are followed by batch normalization and ReLU activation. Following [17], we disable the bias in the convolutional layers.

## D   More Results

### D.1   Tiny ImageNet [8]

Table 1: Results on Tiny ImageNet [8] (in percentage %).

| Method | NMI | ACC | ARI |
|--------|-----|-----|-----|
| DCCM [19] | 22.4 | 10.8 | 3.8 |
| CC [10] | 34.0 | 14.0 | 7.1 |
| TCC | 43.5 | 30.6 | 15.2 |

We show the results on Tiny ImageNet [8] to illustrate the ability of TCC to recognize large numbers of clusters. Only some recent models are compared in Tab. 1 as there are not many articles that include this experiment.

### D.2   More Contrastive Baselines

Table 2: Comparison with [2] on CIFAR-10 (in percentage %).

| Method | NMI | ACC | ARI |
|--------|-----|-----|-----|
| DeepCluster-V2 | 65.2 | 72.7 | 58.9 |
| SwAV [2] | 71.1 | 79.6 | 64.9 |
| TCC | 79.0 | 90.6 | 73.3 |

We notice some pre-text contrastive learning models literally involve a clustering stage as a pre-training module to facilitate downstream tasks such as detection and segmentation, while we simply focus on a model dedicated to clustering. Comparing with them can be also interesting. We show the results with SwAV [2] in Tab. 2. We re-implement these baselines with the same CNN backbone as TCC, and train them at the same resolution as we described in the main body. It can be observed that these baselines obtain on-par performance against some recent methods. Though conceptually including clusters in training, they still mainly focus on better instance-level general representation learning, instead of improving the clustering assignments or preserving more cluster-level semantics.

## D.3 Paired Hypothesis Testing

Table 3: Paired hypothesis test p-values

| | IIC [5] | MMDC [14] | PICA [3] | GATCluster [12] | CC [10] | MoCo baseline [17] | MiCE [17] | IDFD [16] | TCC |
|---|---|---|---|---|---|---|---|---|---|
| IIC [5] | | 0.3213 | 0.0955 | 0.1895 | 0.01 | 0.0313 | 0.0048 | 0.0014 | 0.0005 |
| MMDC [14] | 0.3213 | | 0.1649 | 0.4746 | 0.0105 | 0.1111 | 0.068 | 0.0228 | 0.0075 |
| PICA [3] | 0.0955 | 0.1649 | | 0.024 | 0.0048 | 0.2876 | 0.1219 | 0.0046 | 0.0023 |
| GATCluster [12] | 0.1895 | 0.4746 | 0.024 | | 0.0015 | 0.0551 | 0.0202 | 0.0003 | 0.0002 |
| CC [10] | 0.01 | 0.0105 | 0.0048 | 0.0015 | | 0.0243 | 0.1843 | 0.2563 | 0.0191 |
| MoCo baseline [17] | 0.0313 | 0.1111 | 0.2876 | 0.0551 | 0.0243 | | 0.0199 | 0.0456 | 0.0023 |
| MiCE [17] | 0.0048 | 0.068 | 0.1219 | 0.0202 | 0.1843 | 0.0199 | | 0.121 | 0.0073 |
| IDFD [16] | 0.0014 | 0.0228 | 0.0046 | 0.0003 | 0.2563 | 0.0456 | 0.121 | | 0.0778 |
| TCC | 0.0005 | 0.0075 | 0.0023 | 0.0002 | 0.0191 | 0.0023 | 0.0073 | 0.0778 | |

One of our reviewer suggests showing the performance significance against the existing models. Here we show the p-values of chi-squared test in Tab. 3.