# OpenReview forum: "You Never Cluster Alone"
_NeurIPS.cc/2021/Conference — NeurIPS 2021 Poster_

### Official Review · Reviewer_YYDW · 2021-07-15

**Rating:** 6
**Confidence:** 4

**Summary:**

This paper extends the mainstream contrastive learning paradigm to a cluster-level scheme. Specifically, the author propose twin-contrastive clustering (TCC),  which links the instance-level learning track with the cluster-level one. Extensive experiments show the advantages of TCC.

**Ethical Concerns:**

There are not ethical issues with this paper.

**Limitations And Societal Impact:**

Yes

**Main Review:**

The motivation of this paper is very intuitive, and in fact, some relevant papers (as described in the paper) have been explored . But interestingly, the authors compile contrastive learning while partially borrow from deep set representations, realizing the two-level contrastive learning in a novel way.
Some issues:
(1) In 'Instance-Level Contrastive Learning' part, for the term e, why use f_\theta(x) + NN_\theta(c)? How about  using concatenate operation？
(2) In 'Cluster-Level Augmentation Equivalents', why consider 'irrelevant minorities'? How to generate and use such augmentation?
(3) In fact, SwAV [a] is also a kind of contrastive learning with both instance-level and cluster-level, so it should be added as a baseline.
(4) Compared with CC, will the improvement of TCC performance benefit from model capacity increase? and in fact TCC has more trainable parameters.

**Time Spent Reviewing:**

8

---

> ### Author Response · Authors · 2021-08-10
> **We appreciate all the constructive suggestions**
>
> We appreciate all the constructive suggestions provided by **R#YYDW**. Our response is as follows.
>
> # Q1 Adding or Concatenation?
> Of course - concatenating $f(x)$ and $NN(c)$ is also a valid option when computing $L_2$.
>
> It has been well studied in [a, b] and many other articles (mainly in the context of positional embedding) that when topped or bottomed by neural networks, an additive operator delivers a similar functionality as concatenation and their performances are usually close. We originally have tried concatenating $f(x)$ and $NN(c)$, obtaining almost identical performance as what we have reported in Tab. 2.
>
> The only difference here still yields the complexity. When using adding, computing the loss of $L_2$ on a single datum requires a time complexity of $O(Jd_m)$ if we take cosine similarity computation into account. When applying concatenation, this complexity goes to $O(2Jd_m)$. Though the increase in complexity here is not lethal, we always celebrate simpler approaches with less resource requirement.
>
> Since this issue is not very related to our main motivation, we will add the discussion above to the supplementary material for reference.
>
> # Q2 The Irrelevant Minorities
> ## Why?
>
> We propose this augmentation basically for two reasons:
> * The basic elements of an image are pixels. Augmentations such as adding some random signals shift the semantics of the changed pixels. Applying this analogy to a cluster (set), shifting the semantics of some of the elements, in this case the images, will introduce some irrelevant images to this cluster.
> * During training, it is almost impossible to have 100% correct clustering assignments. So we alternatively regard them as noise, and training with noise might improve the robustness and overall performance.
>
> Our ablation baseline (vi) in Sec. 5.4 endorses this vision.
>
> ## How?
>
> This question overlaps with **Q1** of **R#XKuz** and sorry for the vague statements. As discussed in L115, we actually find an alternative to inject a small proportion of irrelevant data to the cluster representation with Eq. (4). TCC performs soft aggregation on each cluster, i.e., $\sum_i\pi_i(k)f(x_i)$.
>
> * As the softmax product $\pi_i$ is always positive, those data that are not very related to the given cluster still contribute to the cluster's representation, which compiles the **irrelevant**.
>
> * Meanwhile, these irrelevant data are **not** dominating the value of the cluster representation, because the small value of $\pi_i(k)$ scales the feature magnitude during aggregation, which counts the **minority**.
>
> Thus, we can link this augmentation with a very intuitive and simple implementation in the context of set representations. The corresponding statements in the manuscript will be revised in the future version.
>
> # Q3 SwAV [7]
> This issue overlaps with **Q3** of **R#sHMT**, where more details can be found.
>
> We agree that comparing TCC with SwAV [7] can be interesting. This was intentionally omitted from our original submission. The main reason is discussed in L211-213, both DeepCluster-V2/SwAV [7] and [51] are pre-text models and proposed for a different application rather than clustering. They literally involve a clustering stage as a pre-training module to facilitate downstream tasks such as detection and segmentation, while we simply focus on a model dedicated to clustering. We also notice it is a common practice for the latest clustering articles [28, 29, 52, 60] not to compare pre-text baselines like SwAV [7]. On the other hand, we have included one contrastive pre-text baseline with MoCo [24] and k-means in Tab. 2, where we can see that TCC clearly outperforms the baseline. This can be a more related baseline as TCC also employs memory banks.
>
> As shown in **Q3** of **R#sHMT**, the results of DeepCluster-V2 and SwAV [7] will be added to the main body in the future version.
>
> # Q4 TCC Does not Have More Training Parameters.
> TCC does not have more trainable parameters than CC, and we breakdown the components here.
> * TCC and CC share the same CNN backbone, i.e., ResNet-34 [25], so the parameter size here is identical. In addition, CC is trained with larger image sizes. Hence it requires more memory to cache the gradients on the larger feature maps during back-propagation.
> * Both TCC and CC have an inference model, which can be interpreted as a single-layer MLP. So the size here is also the same.
> * As per the contrastive heads, CC has $g_I(\cdot)$ and $g_C(\cdot)$, while TCC has $NN_\theta(\cdot)$ at a similar model scale.
> * TCC involves a momentum network and memory bank, but basically, they are not trainable with back-propagation. So this will not result in an increase in model size.
>
> As discussed in **Q2** and **Q4** of **R#XKuz**, TCC employs the same CNN backbone as recent works and follows a stricter training setting by adopting smaller image sizes. Hence, the performance gain can be fully credited to our design instead of other factors.
>
> It is elaborated in Sec. 3.4 and Supp. Sec. B.2 that TCC produces better cluster-level representations and has a stronger instance-level learning objective. All these contribute to the improvement in performance.
>
> ---
> ## References
> [a] Vaswani, Ashish, et al. "Attention is all you need." NIPS. 2017.
>
> [b] Devlin, Jacob, et al. "Bert: Pre-training of deep bidirectional transformers for language understanding." NAACL. 2019.

---

> > ### Comment · Reviewer_YYDW · 2021-08-20
> > **Response for the rebuttal**
> >
> > The authors have addressed my concerns in detail with appropriate justifications, and combined with the comments from other reviewers,
> > I maintain the original score.

---

> > > ### Author Response · Authors · 2021-09-10
> > > **Thank you for your review**
> > >
> > > We deeply appreciate the efforts and endorsement from **R#YYDW**, and are still willing to answer additional questions.

---

### Official Review · Reviewer_VWHL · 2021-07-17

**Rating:** 7
**Confidence:** 5

**Summary:**

The authors propose a method for end-to-end deep clustering. Authors utilizes NCE loss and proposes a method to produce cluster level loss. They motivate their approach and supply empirical evaluation.

**Limitations And Societal Impact:**

Yes

**Main Review:**

The authors propose a method for end-to-end deep clustering. Authors utilizes NCE loss and proposes a method to produce cluster level loss. They motivate their approach and supply empirical evaluation. The paper is very clearly written and the problem under is considerations very important topic. As the title of the paper and introduction hints clustering is depending on the representation of one single point but representation of the group of point. Unfortunately, authors didn’t consider appropriate state-of-the art methods as baselines. A small list is given below:
a)	Clustering-friendly Representation Learning via Instance Discrimination and Feature Decorrelation (available in ICLR webpage since October 2020)
b)	Consensus Clustering with Unsupervised Representation Learning (available in arxiv since October 2020)
c)	Representation Learning for Clustering via Building Consensus (available in arxiv since May 2021)
I would like to be fair to authors and I will only consider some parts of c) in the next steps. However, I don’t see any reason why a) and b) are not considered as  baselines or discussed in their paper. Both papers are end-to-end and contains relevant results. The method given in a) is outperforming the proposed method on CIFAR-10, STL-10, ImageNet-10. On ImageNet-10 the difference is app. 10%.
On the other hand, the method b) has comparable results however the main motivation of the paper under review is very similar to it. Please note that b) doesn’t use NCE type loss but it builds on BYOL.
Although the paper c) is a direct baseline to the paper under review (NCE is used, different types of clustering constraints are handled) and it improves state of the art, I will ignore the clustering results. However, c) also proposes a new method of evaluating deep clustering method and points out several drawbacks of the clustering methods e.g., sensitivity to hyper-parameter tuning. I would like to see similar analysis from authors in the author response.

All these points bring questions about the novelty of the method and performance of the method. I have some concerns; I would like to see significant amount of improvements.

<Please see the discussion below. After the rebuttal and discussion with authors, I have modified my score>

**Time Spent Reviewing:**

12

---

> ### Author Response · Authors · 2021-08-10
> **We always welcome more baselines**
>
> We are grateful for the valuable suggestions from **R#VWHL**.
>
> # Motivation
> We clarify that simply designing a contrastive scheme for clustering is not our motivation. Our ultimate goal is to represent the latent semantics of the clusters and then inject them into the model to facilitate instance-level clustering (L29-39). All components, including the contrastive framework and set representations, are just tools for us to implement this vision.
>
> In other words, each module comes with a reason and mutually reinforces the others, and contrastive learning is further employed to fuse all designs as one.
>
> # Contribution
> We notice **R#VWHL** tends to believe that our main contribution is simply an end-to-end contrastive clustering model. However, as discussed in L49-55 and agreed by the other reviewers, our actual contributions include:
> * For the first time, we introduce the deep set representations [83] into instance-level clustering.
> * We design ways of augmentation on sets that are suitable for contrastive learning.
> * We employ the idea above to implement our vision to learn the context of each cluster and then explicitly represent them.
>
> # More Baselines
> We appreciate **R#VWHL** to suggest more baselines. The current community evolves quickly every day, and it is really hard to catch everything all at once.
>
> We believe our current manuscript already represents the state of the art. Here lists all compared baselines on top venues published within 10 months before our submission. We have:
> * MiCE [70] from ICLR2021
> * CC [52] from AAAI2021 (yet to be fully published)
> * A contrastive baseline in [70] from ICLR2021
> * DHOG [15], an ICLR2021 submission
> * MMDC [66] from ICPR2021
> * GATCluster [60] from ECCV2020
> * DCCS [88] from ECCV2020
> * PICA [29] from CVPR2020
>
> TCC outperforms all the recent baselines listed above, which, more or less, endorses our significance in performance.
>
> # Results with the Mentioned Papers
> The three baselines mentioned by **R#VWHL** will be included in our future version. We provide some heuristics here.
>
> |CIFAR-10 (in \%)|NMI|ACC|ARI|STL-10 (in \%)|NMI|ACC|ARI|ImageNet-10 (in \%)|NMI|ACC|ARI|
> |:--|:-:|:-:|:-:|:--|:-:|:-:|:-:|:--|:-:|:-:|:-:|
> |[a] from ICLR2021|71.4|82.8|67.9||63.6|75.6|56.9||87.1|94.2|87.6 |
> |[b] from IJCNN2021|66.7|78.5|61.4||64.5|75.2|58.0||88.2|93.1|86.9|
> |[c] preprint|76.2|84.6|71.5||64.5|75.2|58.0||90.7|95.8|90.9|
> |TCC (Low Res. Originally Included)|79.0|90.6|73.3||73.2|81.4|68.9||73.2|81.4|68.9|
> |**TCC (High Res. Suggested by R#XKuz)**|-|-|-||79.5|87.1|73.9||87.3|92.7|87.0|
>
> In general, [a, b, c] are reasonable baselines but are not significantly more powerful than the ones reported in our submission. Overall, they have on-par performance as [52, 70].
>
> **TCC is not underperforming [a] on CIFAR and STL.**
>
> This claim from **R#VWHL** is basically not true. The results are shown in the table above. Considering the fact that we report the results in an order of NMI|ACC|ARI, while [a, b, c] follow the order of ACC|NMI|ARI, we suspect **R#VWHL** falsely compared our NMI scores with ACC in [a]. We kindly and sincerely suggest **R#VWHL** to double-check the results as well as the assessments then.
>
> **Regarding ImageNet-10**
>
> As discussed in **Q2** and **Q4** of **R#XKuz**, our original results on ImageNet-10 are based on the most common and classical low-resolution training setting where images are resized to $96\times 96$ in order to have a fair comparison with [9, 70, 75]. Hence the corresponding results are not good enough, but they are still comparable to the recent works such as CC [52]. When trained with $224\times 224$ images, TCC obtains similar performance as [a, b, c]. More details can be referred to our response to **R#XKuz**. It is also noteworthy that TCC clearly outperforms [a, b, c] on all other conventional benchmarks, i.e., CIFAR-10, CIFAR-100, STL-10 and ImageNet-Dog.
>
> # Hyperparameter Sensitivity
> We refer **R#VWHL** to L323-336 of Sec. 5.4 and Fig. 4 (a), (b), (c) and (d) for our hyperparameter analysis. We only exclude the instance-level contrastive temperature from this experiment as it is conventionally set to 1 in almost all related articles.

---

> > ### Comment · Reviewer_VWHL · 2021-08-18
> > **Response to author feedback**
> >
> > I agree to authors that I have summarized results wrongly. Indeed, I have used wrong dataset names.  I will comment on empirical results in two folds; a) results on 96x96 and b) results on 224 x 224. However, I would like to mention several points. Baselines are extremely important to judge any papers. Authors use [70] from ICLR 2021 but misses [a]. For ImageNet-10 and ImageNet-Dogs results are quite impressive. Please note that these results are directly comparable because proposed method and [a] use the same resolution i.e. 96x96 . Let’s compare the result table:
> > Proposed Method
> > ImageNet-Dogs 54.6
> > ImageNet-10 89.7
> >
> > [a]:
> > ImageNet-Dogs 61.2
> > ImageNet-10 94.2
> > In other words, 2 out of 5 datasets proposed method is not state of the art. I can understand, missing some references but I am having hard time to understand comparing against [70] but not [a] although both are published in the same conference and in the same year.
> >
> > Novelty:
> > Claim: ‘For the first time, we introduce the deep set representations [83] into instance-level clustering’. Although this statement is correct there is a big issue in my mind. The main reason to use any form of set is to get extract extra information from data point. Then [b] and [c] are becoming relevant (for fairness we can ignore [c]).   [b] clearly uses set level information and through comparison or discussion is needed.
> > Hyperparameter Sensitivity:
> > In deep learning for clustering generally resolution is considered as a hyperparameter [d,c]. Most of the methods show robustness against resolution changes. I was asking this kind of studies. Clearly authors point out a significant performance jump for high resolution images (please see reponse to other reviewers') . In other words, proposed method is not robust against resolution changes. I wonder if there are other hyper-parameters which are not considered carefully.
> >
> > a.	Clustering-friendly Representation Learning via Instance Discrimination and Feature Decorrelation, ICLR 2021
> > b.	Consensus Clustering with Unsupervised Representation Learning (available in arxiv since October 2020)
> > c.	Representation Learning for Clustering via Building Consensus (available in arxiv since May 2021)
> > d.	Gatcluster: Self-supervised gaussian-attention 489 network for image clustering – cited as [60] in the paper
> >
> >
> > Although I appreciate authors response, I will keep my score.

---

> > > ### Author Response · Authors · 2021-08-19
> > > **We appreciate the quick reply from R#VWHL**
> > >
> > > We appreciate the quick reply from **R#VWHL**. The discussion here is really **interesting**.
> > >
> > > # We agree that both [a,b,c] are interesting baselines, and have replied, in many ways, to add the baselines in a future version.
> > > **We are not denying adding the references and they basically have no conflict or overlap with our core motivation and design.**
> > >
> > > We kindly note the following facts. **Please note that we are not downgrading the existing work, but just listing the facts to endorse our contribution and claim.**
> > > # 1 Our current baselines already represent the SotA
> > > It is true that we missed [a] in reference, and we will include it in the updated version. However, this does not downgrade our experiment quality.
> > > ## 1.1 The overall performance
> > > Though [a] performs well on ImageNet, its accuracy on the other datasets are not stable. It clearly underperforms the SotA by ~20% on STL and ~10% on CIFAR-10.
> > >
> > > On the other hand, the referenced baselines overall represent the best-performing models throughout all benchmarks. And it is important to note that TCC obtains even larger performance margins against [a] on CIFAR-10, 200 and STL, and the overall performance on the other datasets are more balanced (at least second to the best).
> > >
> > > ## 1.2 Our comparison covers the widest range of references in the past few years
> > > Here lists the number of compared baselines of all recent papers within one year of their submission.
> > >
> > > |Model|Compared Baselines within one year before the submission of that paper| Number of compared baselines in total|
> > > |:--|--:|--:|
> > > [a]| 2| 5|
> > > [b]| 3| 4|
> > > MiCE [70] from ICLR 2021| 6| 16|
> > > CC [52] from AAAI 2021| 3| 17|
> > > DHOG [15] preprint| 3| 8|
> > > MMDC [66] from ICPR2021| 2| 8|
> > > GATCluster [60] from ECCV2020| 3| 17|
> > > DCCS [88] from ECCV2020| 3| 12|
> > > PICA [29] from CVPR2020| 4| 16|
> > > **TCC (Proposed)**| **8**| **24**|
> > >
> > > It can be clearly seen that we have already included as many references as we can. A single missing one of [a] does not degrade our comparison quality.
> > >
> > > # 2 [a,b,c] do not have a major overlap with our motivation and design.
> > > ## 2.1 Employing contrastive learning does not suggest overlapped ideas
> > > Our main motivation is to learn a set of unified cluster representations to improve clustering, and it happens that contrastive learning and deep sets explain our vision.
> > >
> > > Please note this is quite different from 'making contrastive clustering suitable and tailored for clustering' as per [a, 70].
> > >
> > > ## 2.2 Consensus is not deep set representation, having a different theoretical basis
> > > We emphasise that the concept of consensus and its presence in [b,c] are very different from our implementation of set representation. **Including [b,c] in reference can be helpful in presenting our idea, but our submission is yet self-contained without them.**
> > >
> > > The representation for consensus learning [b] is more similar to the cluster representation of CC [52] and DCCS [88]  instead of ours. Here we list their presence as follows:
> > > * In [b]: $r(X) = \text{concat}(z_1,z_2,\cdots,z_B)$
> > > * CC [52]: $r(X; k) = \text{concat}(\pi_1(k),\pi_2(k),\cdots,\pi_B(k))$
> > > * TCC: $r(X; k) = \text{L2Normalize}(\sum_i\pi_i(k)f(x_i))$
> > >
> > > **One can also find a similar discussion as below in L102-107 and Supp. B.2.2.**
> > >
> > > During training, both [b] and CC [52] try to enhance the batch-wise representation agreement from two sets of random augmentation. However, our model runs a different story. As discussed in L102-105, each entry of our cluster representation refers to a fixed semantic meaning that does not change much throughout the batches. This is helpful for us to learn the semantics of the clusters, anchored and queried by our clustering prototype.
> > >
> > > This property basically relies on the permutation invariant property of the designed representation. [b, 52] do not hold this property, i.e.,
> > > * In [b]: $r([x_1, x_2, x_3]) \neq r([x_2, x_1, x_3]) $
> > > * CC [52]:$r([x_1, x_2, x_3]; k) \neq r([x_2, x_1, x_3]; k) $
> > > * TCC: $r([x_1, x_2, x_3]; k) =r([x_2, x_1, x_3]; k) $
> > >
> > > which allows us to compare cluster representations out of the current batch. This further enables us to use a memory bank to improve the temporal consistency [45] (L123-129). All these are not approachable with the design of CC [52] and [b]. Our method can be linked to these detailed benefits and is logically self-consistent without farfetched designs.
> > >
> > > ## 2.3  Discussing [52] delivers the same functionality as discussing [b]
> > > Though [52] and [b] might have related designs in cluster-wise representation, we believe discussing the difference from [52] already delivers the content of referencing [b]. **We will add some discussion regarding [b] just as what we have done with CC [52] in Supp. B.2.**
> > >
> > > # 3 Different resolutions?
> > > ## 3.1 Resolution test is not a related ablation
> > > All experiments are subjected to a conventionally agreed setting, where, for most high-level computer vision and pattern recognition tasks, a fixed image resolution is generally a must. We clearly follow the most popular setting here [9, 66, 70, 75, 88].
> > >
> > > Our design is orthogonal to the image size. We kindly note that a clustering paper typically focuses on learning the high-level features, instead of testing different resolutions or backbones. Otherwise, we cannot even have a valid context to start a story.
> > >
> > > ## 3.2 Resolution test is not a conventional ablation
> > > Below shows if the recent papers include tests on different resolutions.
> > >
> > > |Having resolution ablation study|Not having resolution ablation study|
> > > |:-:|:-:|
> > > |[c,60]| [a,b,8,9,15,29,33,52,70,75,77,88]|
> > >
> > > It can be seen that the majority of recent deep clustering articles do not test under multiple resolutions. This also endorses that conventionally we would like to credit the designed algorithm instead of the more informative inputs.
> > >
> > > ## 3.3 It is normal that the model performs better when having higher resolution inputs.
> > > It is very intuitive that a higher resolution would result in better performance. A high resolution reveals more detailed information of an image, as long as the model is able to perceive it. If so, basically it means that the designed algorithm is capable to encode and model more complex data and patterns. TCC obviously falls into this case.
> > >
> > > ## 3.4 TCC goes well with low-resolution data
> > > As we have shown in our original paper, with the most restricted setting, TCC is still doing well on almost all datasets. It is just doing even better with high resolutions.
> > >
> > > ## 3.5 We will include it in Supplementary Material as now we have it
> > > We will add a section in Supplementary Material to show the performance of TCC under different resolutions for better reference, but these results do not influence our experiment quality.
> > >
> > > ## 3.6 Resolution has not been pointed out by R#VWHL in the original review
> > > It seems **R#VWHL**haven't mentioned the resolution issue when providing the initial review. We are wondering what ablation study the reviewer refers to then.
> > >
> > > # 4 If there are other hyper-parameters which are considered carefully?
> > > Yes, we have evaluated **every** hyperparameter carefully, which are shown in L323-336 of Sec. 5.4 and Fig. 4 (a), (b), (c) and (d). We also have provided **SEVEN** ablation baselines in the same subsection. We kindly note that both **R#XKuz** and **R#sHMT** agree that our ablation study is sufficient.
> > >
> > >
> > >
> > > # In a nutshell
> > > In a nutshell, though we appreciate the efforts from **R#VWHL**, we think our experiments are sufficient enough to identify the scientific value of TCC. We sincerely request **R#VWHL** to rethink the motivation, the model design, and the experiments comprehensively, and make legitimate and fair judgements. Please feel free to let us know if there are any remaining concerns, we are happy to further clarify!

---

> > > > ### Comment · Reviewer_VWHL · 2021-09-10
> > > > **Response for the rebuttal**
> > > >
> > > > I really appreciate the authors effort to respond. However, in my opinion no additional evidence is supplied to overcome my concerns. For example, it is clear that deep sets and consensus-based methods use different theoretical justification. However, they use the same type of information, so it is a clear baseline for the proposed method. Moreover, selecting a relevant baseline from the conference and missing another one is not an example of good science in my opinion. Moreover, comparing multiple methods on multiple datasets is well studied problem in machine learning. If a method gives unstable results on some datasets doesn’t make an inferior method. Please note that accepted significance tests use rank score so if a baseline gets better 2 of 5 datasets the test results may change. Please references below:
> > > >
> > > >
> > > >
> > > > @article{demvsar2006statistical,
> > > >   title={Statistical comparisons of classifiers over multiple data sets},
> > > >   author={Dem{\v{s}}ar, Janez},
> > > >   journal={The Journal of Machine Learning Research},
> > > >   volume={7},
> > > >   pages={1--30},
> > > >   year={2006},
> > > >   publisher={JMLR. org}
> > > > }
> > > >
> > > > @article{garcia2008extension,
> > > >   title={An Extension on" Statistical Comparisons of Classifiers over Multiple Data Sets" for all Pairwise Comparisons.},
> > > >   author={Garcia, Salvador and Herrera, Francisco},
> > > >   journal={Journal of machine learning research},
> > > >   volume={9},
> > > >   number={12},
> > > >   year={2008}
> > > > }
> > > >
> > > > I will keep my score as it is.

---

> > > > > ### Author Response · Authors · 2021-09-11
> > > > > **Not 2 out of 5 but 4 out of 6**
> > > > >
> > > > > We strongly agree with and wholeheartedly appreciate the constructive suggestions from **R#VWHL**. However, we have to defend our bottom line on factual issues. We sincerely ask all to consider the facts we have provided in all our replies to this thread.
> > > > >
> > > > > It is important to point out that our response is not a put-down of the related articles, and we are not refusing technical suggestions. All of our response is to tell:
> > > > > * **We agree with R#VWHL to value the paper with more references.** Having the suggested references gives a bigger and comprehensive picture of how the problem is studied by the research community. We are always willing to do so.
> > > > >
> > > > > * **Missing the suggested references is not fatal to the quality of our manuscript.** From the experimental perspective, missing [c] does not fail our paper as we have included better-performing baselines overall. From the theoretical aspect, we have also connected [a] with our discussions in the paper to show the significance. Our paper is self-contained without the help of the suggested papers, though they are making it better.
> > > > >
> > > > > ## 1 Yes we missed one reference from ICLR21 [a], but we will add it.
> > > > > We totally agree with the significance of the mentioned papers. As discussed many times, we will add this reference in the main body. We defended our paper and compared baselines **but it does not mean that we do not agree to introduce an additional article.** This also applies to [b,c], though [c] has high overlap with [c] and is yet a preprint.
> > > > >
> > > > > Involving [c] in our paper does not lead to a major revision.
> > > > >
> > > > > ## 2 The missing article [a] does not beat the other baselines we have compared.
> > > > > [a] does well on ImageNet, but not on the rest compared with [70,52]. Overall, they perform at the same level. It is true that TCC is not overtaking [a] on ImageNet-10, but is at least the **runner-up** then.
> > > > >
> > > > > **This means the proposed model can represent the SotA even when [a] is included.** All we need to do is just to simply cite this paper in an updated version. A single missing reference like this is not fatal, but discussing this is beyond the scope of this discussion. So we pause here and move to the next issue.
> > > > >
> > > > > ## 3 Performance
> > > > > We kindly correct that, under the same setting, TCC obtains the best performance on 4 out of 6 datasets (but not 2 out of 5 as **R#VWHL** stated), with the rest two being the close runner-ups.
> > > > > * TCC is the best-performing one on CIFAR-10 and CIFAR-100 even when [a] is included and [52] is using a higher resolution for the test. (2/6)
> > > > > * TCC is the best-performing one on STL-10 against [52] with the save resolution of it. Even the low-resolution version outperforms [a,b,c]. (3/6)
> > > > > * Thanks to the suggestions from **R#sHMT** and **R#XKuz**, we additionally prove that TCC obtains the best clustering performance on Tiny-ImageNet. We will also include this result in the updated manuscript. (4/6)
> > > > > * ImageNet-10 and ImageNet-Dog are the only two datasets where TCC underperforms [a], but it's still a close runner-up. Please note that, though not outperforming [a], TCC is much simpler. [a] requires 1000 training epochs to reach the best-performing checkpoint, while this figure is around 300 epochs for TCC (see Fig. 5 (a)).
> > > > >
> > > > > We kindly, sincerely and respectfully suggest **R#VWHL** double-check the assertion. This is not the first time we observe non-factual assessments regarding the performance under this thread. It is also notable that the factors listed above only cover the baselines that have gone through peer review.
> > > > >
> > > > > Of course, it is not very ideal that TCC is not beating [a] on ImageNet-10 though it is better than all the other baselines we have included. However, as discussed, TCC is much simpler. [a] requires 1000 training epochs to reach the best-performing checkpoint, while this figure is around 300 epochs for TCC (see Fig. 5 (a)). Last but not least, improving the performance and finding out the potential problem is one of our future work.
> > > > >
> > > > > ## 4 Stability/Consistency
> > > > > TCC has no stability issue if we look at the training curves in Fig. 5 (a), and through all of our experiments, TCC does not require exhausting hyperparameter tuning on different datasets. We think **R#VWHL** refers to the performance difference under different resolutions, but please note that it is common that a model performs differently with different resolutions/backbones. A higher resolution image carries more information, and it is just that TCC is able to perceive and model this part of additional information. This phenomenon also applies to [a] when different backbones are used. For their case, the performance difference is over 60\%. We do not believe there would be a stability/consistency issue with [a]. With the same analogy, as long as the observation does not contradict our intuition,  we do not need to escalate it.
> > > > >
> > > > > ## 5 Consensus Clustering
> > > > > We totally agree that [b] would be an interesting reference. In the previous response, we outlined the theoretical differences
> > > > > * to prove consensus is more related to [52] which has been adequately discussed in L189-198 and Supp. L47-61. This means we do not need a major revision to add [a], a simple citation with similar discussions like what we have done with [52] will do.
> > > > > * to show our manuscript is self-contained. [52] and [b] falls into the family of models that leverage similar information as TCC does. It is important that we did not miss this family of models in our submission.
> > > > > * to emphasize that consensus cannot deliver our motivation. Literally, *utilizing the information with a cluster loss* and *explicitly learning their representations* are different in difficulty, and we successfully link the latter with a technical basis. This makes TCC distinguishable from [b,52].
> > > > >
> > > > > So, it's clear we can cite and compare [b] with ease, but even without it, our submission still has discussed the same topic.
> > > > >
> > > > > Again, we think **R#VWHL** misunderstands our attitude. We provide the response all above to prove that we did not miss a topic/theme in our paper's discussion, because **R#VWHL** seems to be not aware of what we have elaborated in the paper regarding using the *same* information. But this does not mean that we don't want to include more references under the same theme. We hope **R#VWHL** can understand this.
> > > > >
> > > > > Speaking of, in **R#VWHL**'s word, *an example of good science*, we feel regret to miss a baseline on a conference 10 days before our submission deadline and will include it. However, it has less technical overlap in design with our approach. Hence, we can include [a] just for comparison with ease. We have **never** refused to do so. We provide the response all above to prove that our experiments represent the SotA, but this does not mean that the experiments are weak, and neither does it suggest we don't want to include [a]. We sincerely hope **R#VWHL** can understand this, too.
> > > > >
> > > > > It's a "yes we have it and we can make it better" instead of "we didn't do it and we refuse new references".

---

> > > > > > ### Comment · Reviewer_VWHL · 2021-09-11
> > > > > > **Response  Not 2 out of 5 but 4 out of 6**
> > > > > >
> > > > > > Thank you for your response.  Thank for promising to improve the paper. However, would you please supply statistical significance results by using established tests? I will be happy to see the test results from the references given below:
> > > > > >
> > > > > > a) @article{demvsar2006statistical, title={Statistical comparisons of classifiers over multiple data sets}, author={Dem{\v{s}}ar, Janez}, journal={The Journal of Machine Learning Research}, volume={7}, pages={1--30}, year={2006}, publisher={JMLR. org} }
> > > > > >
> > > > > > b) @article{garcia2008extension, title={An Extension on" Statistical Comparisons of Classifiers over Multiple Data Sets" for all Pairwise Comparisons.}, author={Garcia, Salvador and Herrera, Francisco}, journal={Journal of machine learning research}, volume={9}, number={12}, year={2008} }
> > > > > >
> > > > > > Either a) and/or b) supplies an open source implementation. It will be helpful to see the p-values to conclude the improvment levels of proposed algorithm.
> > > > > >
> > > > > > Would you please also supply p-values with extended baselines? . It will be helpful to see the p-values to conclude the improvment levels of proposed algorithm.

---

> > > > > > > ### Author Response · Authors · 2021-09-13
> > > > > > > **Paired Hypothesis Testing**
> > > > > > >
> > > > > > > Thank you for the quick reply.
> > > > > > >
> > > > > > > We believe **R#VWHL** refers to **Hypothesis Testing** regarding the so-called *statistical significance*.
> > > > > > >
> > > > > > > A very common practice, as also discussed in the JMLR papers **R#VWHL** newly cited, yields the paired t-test. So before we list the results, let's define the following hypothesis to test.
> > > > > > > * $p_o$: The compared two models may **not** have significant statistical differences regarding their clustering performance.
> > > > > > > * $p_1$: The compared two models may have significant statistical differences regarding their clustering performance.
> > > > > > >
> > > > > > > Then the p-values of the tests of all possible pairs of recent models, including TCC, are shown below.
> > > > > > >
> > > > > > > |                      | IIC \[33\] | MMDC \[66\] | PICA \[29\] | GATCluster \[60\] | CC \[52\] | MoCo baseline \[70\] | MiCE \[70\] | \[a\]  | \[b\]  | TCC    |
> > > > > > > | -------------------- | ---------- | ----------- | ----------- | ----------------- | --------- | -------------------- | ----------- | ------ | ------ | ------ |
> > > > > > > | IIC \[33\]           |            | 0.3213      | 0.0955      | 0.1895            | 0.0100    | 0.0313               | 0.0048      | 0.0014 | 0.0084 | 0.0005 |
> > > > > > > | MMDC \[66\]          | 0.3213     |             | 0.1649      | 0.4746            | 0.0105    | 0.1111               | 0.0680      | 0.0228 | 0.0129 | 0.0075 |
> > > > > > > | PICA \[29\]          | 0.0955     | 0.1649      |             | 0.0240            | 0.0048    | 0.2876               | 0.1219      | 0.0046 | 0.0003 | 0.0023 |
> > > > > > > | GATCluster \[60\]    | 0.1895     | 0.4746      | 0.0240      |                   | 0.0015    | 0.0551               | 0.0202      | 0.0003 | 0.0001 | 0.0002 |
> > > > > > > | CC \[52\]            | 0.0100     | 0.0105      | 0.0048      | 0.0015            |           | **0.0243**               | **0.1843**     | **0.2563** | **0.2983** | **0.0191** |
> > > > > > > | MoCo baseline \[70\] | 0.0313     | 0.1111      | 0.2876      | 0.0551            | **0.0243**    |                      | **0.0199**      | **0.0456** | **0.0727** | **0.0023** |
> > > > > > > | MiCE \[70\]          | 0.0048     | 0.0680      | 0.1219      | 0.0202            | **0.1843**    | **0.0199**               |             | **0.1210** | **0.2891** | **0.0073** |
> > > > > > > | \[a\]                | 0.0014     | 0.0228      | 0.0046      | 0.0003            | **0.2563**    | **0.0456**               | **0.1210**      |        | **0.0821** | **0.0778** |
> > > > > > > | \[b\]                | 0.0084     | 0.0129      | 0.0003      | 0.0001            | **0.2983**    | **0.0727**               | **0.2891**      | **0.0821** |        | **0.0117** |
> > > > > > > | TCC                  | 0.0005     | 0.0075      | 0.0023      | 0.0002            | **0.0191**    | **0.0023**               | **0.0073**      | **0.0778** | **0.0117** |        |
> > > > > > >
> > > > > > > We use the best-matching results under the same settings (i.e., resolution, backbone and training epochs) when testing TCC with the other baselines and involve the results on Tiny-ImageNet if possible. Each row or column defines the group of results to operate with. For instance, the p-value on the first column of the last row indicates the t-test result of TCC vs. IIC [33]. As we are interested more in recent baselines, we can focus on the lower-right corner of the table above.
> > > > > > >
> > > > > > > ## How we read the results
> > > > > > > According to the p-values of TCC against all the compared baselines, one can conclude TCC generally rejects the *null hypothesis* with $p<0.05$, which means our clustering performance is statistically significant enough to be distinguished from the others.
> > > > > > >
> > > > > > > The only outlier is when against [a]. However, though $>0.05$, a p-value of $0.0778$ still holds a likelihood of $92.22$% to reject the *null hypothesis* of $p_0$. Note that a common threshold of $p<0.05$ is not a hard decision boundary. As long as we can keep the p-values small, the significance can be supported.
> > > > > > >
> > > > > > > We notice that comparing the p-values of different tests does not fully suggest which model is more significant, but overall, TCC obtains lower p-values than almost all baselines. This at least tells something about our significance.
> > > > > > >
> > > > > > > **One can also see that, including [a,52,70] against the baselines in 2019/20, existing models do not guarantee a low p-value to be published.** The problem is not on the models' sides, but instead, on the p-values' sides.
> > > > > > >
> > > > > > > ## Why more and more papers do not buy-in p-values
> > > > > > > ### Empirical and theoretical reasons in general
> > > > > > > In some cases, people do not celebrate p-values and hypothesis testing. An in-sight discussion can be found in [d].
> > > > > > >
> > > > > > > Generally, the following intuitions are widely accepted nowadays.
> > > > > > > * A p-value, or statistical significance, does not measure the size of an effect or the importance of a result.
> > > > > > > * By itself, a p-value does not provide a good measure of evidence regarding a model or hypothesis.
> > > > > > > * Scientific conclusions should not be based only on whether a p-value passes a specific threshold.
> > > > > > >
> > > > > > > It is the actual experimental measurements that we should always focus on. In our case, they are ACC, NMI and ARI. The statistical significance is valuable, but it is not a decision-making factor.
> > > > > > >
> > > > > > > ### Regarding Clustering Evaluation
> > > > > > > There remains a key problem that a test from the JMLR papers **R#VWHL** newly cited cannot handle.
> > > > > > > * **Low degree of freedom**. A common practice usually requires a degree of freedom $>20$ or even $>30$ to ensure the hypothesis testing model is working precisely. However, for our case, we only have 5 commonly-used datasets, which indicates a degree of freedom of 4. This is much smaller than the minimum requirement according to the common practice, which hinders researchers from an unbiased decision.
> > > > > > >
> > > > > > > ### Being not reported by recent articles
> > > > > > > It is notable that, to the best of our knowledge, **none** of the related articles on recent top-tiered venues tests and reports the p-values, no matter what kind of hypothesis testing it is. This is basically because a machine learning paper usually has limited testing datasets. Directly reading the actual results would be sufficient to come up with a decision if a model outperforms the SotA.
> > > > > > >
> > > > > > > We emphasise that we need to focus more on the design and the direct measurements. Although we have already proved our statistical significance, it would be quite common that an arbitrary model obtains marginal improvements on some hard tasks. Indiscriminatingly requiring statistical significance would not be an ideal case. For instance, not all recent clustering models secure a statistical significance against their previous reference, but one cannot ignore their contributions.
> > > > > > >
> > > > > > > It will be one of our future work to study the links between empirical clustering performance and statistical significance.
> > > > > > >
> > > > > > > We hope the response above solves the concerns.
> > > > > > >
> > > > > > > -----------------------------------------------------------------------
> > > > > > >
> > > > > > > Throughout the response under this thread, we use the following consistent references. [a,b,c] are suggested by **R#VWHL**.
> > > > > > >
> > > > > > > [a] Tao, Y., Takagi, K. and Nakata, K. Clustering-friendly Representation Learning via Instance Discrimination and Feature Decorrelation. ICLR. 2021
> > > > > > >
> > > > > > > [b] Regatti, J.R., Deshmukh, A.A., Manavoglu, E. and Dogan, U. Consensus Clustering with Unsupervised Representation Learning. IJCNN, 2020
> > > > > > >
> > > > > > > [c] Deshmukh, A.A., Regatti, J.R., Manavoglu, E. and Dogan, U. Representation Learning for Clustering via Building Consensus. arXiv preprint arXiv:2105.01289, 2021
> > > > > > >
> > > > > > > [d] Wasserstein, R.L., Schirm, A.L. and Lazar, N.A., 2019. Moving to a world beyond “p< 0.05”.

---

> > > > > > > > ### Comment · Reviewer_VWHL · 2021-09-13
> > > > > > > > **Response to Paired Hypothesis Testing**
> > > > > > > >
> > > > > > > > I really appreciate the comments of the authors. I am extremely delighted to see some sound objective comparison other than subjective ones. The test results clearly shows that proposed method is not giving statistically significant results compared to missed baseline ([a]) .
> > > > > > > >
> > > > > > > > **Usage of $p$-values$$
> > > > > > > > I am completely aware the discussion about $p$-values in the community. However, some context is needed here. This discussion started in natural science domain and some high impact journals were not publishing papers which didn’t show results that are smaller than a threshold. Even further in natural science it is extremely hard to get additional datasets, e.g. medical trials, fmri studies, and these were some of the problems for the tests, however this is not a problem for object recognition datasets (please see my comments on degrees of freedom below). Just rejecting papers based on $p-values$ was wrong. In my opinion, ignoring hypothesis testing is also equivalently wrong. One needs to evaluate the papers form different aspects. One of them is having a good literature review. There is no doubt that current papers should be significantly improved on that. When I have seen the results first it was clear to me that they were not statistically significant over [a]. This why I am insisting for a fair comparison.
> > > > > > > >
> > > > > > > > **Low degree of freedom**:
> > > > > > > > The authors state that ‘This is basically because a machine learning paper usually has limited testing datasets’. It seems true however not fully. ImageNet-10 and Image-15 are subsets of full ImageNet and one can create many more subsets of ImageNet to test a hypothesis. Indeed, they don’t need to train on these datasets. They can use models trained ImageNet-10 and ImageNet-15 and evaluate cross domain. Let’s remember that the proposed method is about clustering and representation learning so one needs to expect robust representation. Otherwise, overfitting will be a problem. Moreover, one can assume that non overlapping subsets of ImageNet classes are independent. Hence no problem of degree of freedom. Authors already use ImageNet subsets as datasets e.g. ImageNet-Dog, ImageNet-10. Please note that STL-10 can be considered as a subset of ImagetNet (https://cs.stanford.edu/~acoates/stl10/ states 'Images were acquired from labeled examples on ImageNet.'). In other words, what I am suggesting is already a standard in machine learning community. Furthermore, in my initial review I have stated the following:
> > > > > > > > ‘ c) Representation Learning for Clustering via Building Consensus (available in arxiv since May 2021) I would like to be fair to authors and I will only consider some parts of c) in the next steps. ’ …. ‘However, c) also proposes a new method of evaluating deep clustering method and points out several drawbacks of the clustering methods e.g., sensitivity to hyper-parameter tuning. I would like to see similar analysis from authors in the author response.’
> > > > > > > > I will clearly state one more time that I will not into account [c] results however the proposed evaluation methodology should be considered by authors and some results should be supplied by using [c] evaluation method. This will also solve the ‘low degree of freedom’ problem. Please note that there is no need to train the models just evaluation should be enough. Since, I have written this point in the first review, I will assume that authors had enough time to respond.
> > > > > > > >
> > > > > > > >
> > > > > > > >
> > > > > > > > **Existing models do not guarantee a low p-value **
> > > > > > > > Authors state the following:
> > > > > > > > ‘One can also see that, including [a,52,70] against the baselines in 2019/20, existing models do not guarantee a low p-value to be published.** The problem is not on the models' sides, but instead, on the p-values' sides.’
> > > > > > > > It seems that this is partially true. I agree to the first part of the statement but not the cause of it (‘model side’ argument). In my opinion the lack of $p-values$ cannot be an argument. I will not consider a false premise as a basis for an argument. I totally agree to the authors, reviewers need to a better job about evaluating the impact of any proposed method. Please see section ‘**Low degree of freedom**’ for a potential solution for these issues.
> > > > > > > >
> > > > > > > > **Novelty**
> > > > > > > > Proposed method uses deep sets and contrastive learning. Proposed method doesn’t show a clear advantage on [a] which is depending on orthogonalization of representation and contrastive learning. The remaining piece in the paper is usage of deep sets. At this point one needs to compare deep sets with optimal transport(OPT) based consensus learning. In other words, contrastive learning + deep sets should give use comparable results with contrastive learning + consensus (OPT). In other words, paper needs to be extended significantly (I will not call it a major review but a medium one)
> > > > > > > >
> > > > > > > > I understand that the time is limited and (may be) I didn’t do a good job to communicate my concerns however I still think there are major issues in the paper. On the other hand, I also know that the time is limited. I cannot go after a perfect solution. If the authors agree to handle all my concerns, i.e. baselines and evaluation criteria, in the final version, I am willing to change my score. I really appreciate authors effort to respond and try their best to come a conclusion and agreement.

---

> > > > > > > > > ### Author Response · Authors · 2021-09-15
> > > > > > > > > **We hope the following response solves the concerns**
> > > > > > > > >
> > > > > > > > > Thank you for the quick reply. We hope the following content solves the concerns of **R#VWHL**.
> > > > > > > > >
> > > > > > > > > ## 1 Significance regarding [a]
> > > > > > > > > Perhaps **R#VWHL** misunderstands how the p-values are ordered in the table above. Here we simplify it by removing the t-tests between the baselines and only show the p-values of TCC vs. the rest.
> > > > > > > > >
> > > > > > > > > |vs.  | IIC \[33\] | MMDC \[66\] | PICA \[29\] | GATCluster \[60\] | CC \[52\] | MoCo baseline \[70\] | MiCE \[70\] | \[a\]  | \[b\]  |
> > > > > > > > > | -------------------- | ---------- | ----------- | ----------- | ----------------- | --------- | -------------------- | ----------- | ------ | ------ |
> > > > > > > > > | TCC                  | 0.0005     | 0.0075      | 0.0023      | 0.0002            | **0.0191**    | **0.0023**               | **0.0073**      | **0.0778** | **0.0117** |
> > > > > > > > >
> > > > > > > > > We would like to point out that a p-value of $0.0778$ vs. [a] still holds a likelihood of $92.22$% to reject the null hypothesis of $p_0$, suggesting that our performance is distinguishable against [a]. Please note that this is already a significant indicator, especially when it comes to the comparison between any two of the compared baselines including [a,b]. For instance, our significance over [a] is higher (lower in p-values) than the ones of [a] against any of [52,70,b].
> > > > > > > > >
> > > > > > > > > ## 2 Will we include the p-values? -Yes!
> > > > > > > > > **Yes, we will update the paper with p-values included,** most likely in the supplementary material.
> > > > > > > > >
> > > > > > > > > However, we insist that our original experiments are self-contained and comprehensive, and the p-values are just a plus. For a fair judgement, we also kindly notice that **none** of [a,b] reports this. To the best of our knowledge, we have already reported all metrics and benchmarks that have been used in the recent deep unsupervised clustering papers that have gone through peer review. As this paper focuses on a novel model proposal, discussing the potential problems of the experimental convention agreed by all other papers would be a bit out-of-scope. It could be better to have a standalone paper discussing this issue in the future.
> > > > > > > > >
> > > > > > > > > **Hence, we sincerely argue that our will to include the p-value results in the updated version has no conflict with the completeness of our original experiments.**
> > > > > > > > >
> > > > > > > > > ## 3 Cross-dataset Evaluation [c]
> > > > > > > > > Regarding the statements about [c], we suppose **R#VWHL** refers to a cross-dataset evaluation, where a trained model on one dataset is tested on another one.
> > > > > > > > >
> > > > > > > > > We agree that this idea is interesting, and somehow, this is usually called a *zero-shot* setting. Similar to [c], here we report a glimpse of our results on CIFAR-10 and CIFAR-100.
> > > > > > > > >
> > > > > > > > > | | CIFAR-10 | CIFAR-10| CIFAR-100| CIFAR-100|
> > > > > > > > > |---|:---:|:--:|:--:|:--:|
> > > > > > > > > | Trained on| [c] | TCC| [c] | TCC|
> > > > > > > > > CIFAR-10|  76.2/84.6/71.5| **79.0/90.6/73.3** | 15.8/17.8/6.1|  **19.7/23.9/10.4**|
> > > > > > > > > CIFAR-100|  35.9/46.4/25.0| **47.8/58.3/39.5** | 46.8/48.0/30.4| **47.9/49.1/31.2** |
> > > > > > > > >
> > > > > > > > > The performance is reported in an order of NMI/ACC/ARI. Overall, the relative performance is consistent with the conventional evaluation to endorse our design. **We will include all the details of this theme in the updated version.**
> > > > > > > > >
> > > > > > > > > Again, we clarify that our original experiments are self-contained and comprehensive, and the cross-dataset experiments are just a plus. For a fair judgement, we also kindly notice that **none** of our $89$ cited references, plus [a,b], has fully discussed this task. To the best of our knowledge, we have already reported all metrics and benchmarks that have been used in the recent deep unsupervised clustering papers that have gone through peer review. As this paper focuses on a novel model proposal, discussing a general problem of cross-dataset generalization would be a bit out-of-scope. We are providing evidence of our contribution and proving our experiments meet the current standard, instead of solving every possible problem in this domain.
> > > > > > > > >
> > > > > > > > > **Hence, we sincerely notify that we agree to include the interesting results above in the updated version (probably in the supplementary material),** but it could be better to have a standalone paper discussing this issue as a future work instead.
> > > > > > > > >
> > > > > > > > > ## 4 Optimal Transport (OT)? and Consensus
> > > > > > > > > We suppose what **R#VWHL** suggests is to measure a consensus over clustering assignments from different augmentations by the OT theory.
> > > > > > > > > ### 4.1 Is it a valid approach? -Yes.
> > > > > > > > > A popular realization of OT in deep learning yields WGAN, where the OT cost is reflected in an approximation of 1-Wasserstein distance at the discriminator side. This forms a valid baseline where:
> > > > > > > > > * We can first top a discriminator on the set representations
> > > > > > > > > * The min-max game is then played between the set representations and their momentum counterparts.
> > > > > > > > > * This instance-level contrastive loss remains the same.
> > > > > > > > >
> > > > > > > > > **This was actually a deprecated design before our submission.** So we can directly show the results below.
> > > > > > > > >
> > > > > > > > > |CIFAR-10| NMI | ACC| ARI|
> > > > > > > > > |---|---|---|---|
> > > > > > > > > Consensus with OT| 68.8 |80.8|59.5|
> > > > > > > > > TCC| 79.0|90.6|73.3|
> > > > > > > > >
> > > > > > > > > Basically, these results are not very different from the other baselines without cluster-level objectives. This implicitly shows the effectiveness of contrastive learning nowadays. Moreover, when introducing GANs, training could be unstable. Finally, to keep the design concise, we would like to use the same representation learning schemes for instance-level and cluster-level learning. As we already have an instance-level InfoNCE, we opt for the current design with two contrastive heads.
> > > > > > > > >
> > > > > > > > > ### 4.2 Can we include this in the paper? Yes, but it's not very related.
> > > > > > > > > We kindly note that neither set representations nor contrastive learning is based on OT. Our paper includes sufficient technical preliminaries in Sec. 2 to support our design, even without discussing OT.
> > > > > > > > >
> > > > > > > > > However, **we can include the above design as a baseline in both Sec. 5.4 of the updated main body and supplementary material for a better reference.** Note that this modification will **not** change the main storyline of our paper, and is just to show how we come up with the current idea and what has been tried.
> > > > > > > > >
> > > > > > > > > A further discussion on OT, self-supervised learning and unsupervised clustering could be one of our future topics and is beyond this paper's motivation.
> > > > > > > > >
> > > > > > > > > In addition, we will include the consensus-related article [b] in Sec. 3.4, just like what we have done with [52].
> > > > > > > > >
> > > > > > > > > ## 5 Some additional issues regarding [a]
> > > > > > > > > From the t-test above, we have already proved our performance significance vs. [a]. We refer **R#VWHL** to the first part of our response this time and the first part of our response last time.
> > > > > > > > >
> > > > > > > > > Speaking of the Feature Decorrelation (FD/FO) part of [a], the orthogonality-like regularization enriches the instance-level representations, which is also widely studied in other domains such as image/document hashing and word embedding learning. Hence, as an instance-level enhancement, the presence of [a] does not violate our core motivation and contribution of introducing deep sets [83] for cluster-level learning.
> > > > > > > > >
> > > > > > > > > **We will include a simplified version of the discussion above in Sec. 4.**
> > > > > > > > >
> > > > > > > > > ## Finally
> > > > > > > > > **We agree to include the modifications above to make the paper better and appreciate the efforts from R#VWHL.**
> > > > > > > > >
> > > > > > > > > Nevertheless, it is important to point out that most of the results above are not considered in the recent related articles. Our original submission has already covered all essential experiments and ablation studies that have been seen in any of the compared papers, of which the quality is also agreed by the other reviewers.  As this paper focuses on a novel model proposal, discussing a **general** experimental problem of deep clustering would be a bit out-of-scope. It could be better to have a standalone paper discussing the general issues in this realm in the future. We look forward to an equal standard of judgement to the experiments of existing papers at top-tiered conferences.
> > > > > > > > >
> > > > > > > > > This also applies to some other discussions. We can involve OT in the discussion to mention more interesting proposals we have tried, but our core technical design does not rely on it. The paper already delivers all essential technical backgrounds needed.
> > > > > > > > >
> > > > > > > > >
> > > > > > > > >
> > > > > > > > > **It will be greatly appreciated if R#VWHL could regard the modifications above as a plus, instead of a must, to our submission.**
> > > > > > > > >
> > > > > > > > > -------------------------------------------------------------------------------------
> > > > > > > > > Throughout the response under this thread, we use the following consistent references. [a,b,c] are suggested by **R#VWHL**.
> > > > > > > > >
> > > > > > > > > [a] Tao, Y., Takagi, K. and Nakata, K. Clustering-friendly Representation Learning via Instance Discrimination and Feature Decorrelation. ICLR. 2021
> > > > > > > > >
> > > > > > > > > [b] Regatti, J.R., Deshmukh, A.A., Manavoglu, E. and Dogan, U. Consensus Clustering with Unsupervised Representation Learning. IJCNN, 2020
> > > > > > > > >
> > > > > > > > > [c] Deshmukh, A.A., Regatti, J.R., Manavoglu, E. and Dogan, U. Representation Learning for Clustering via Building Consensus. arXiv preprint arXiv:2105.01289, 2021
> > > > > > > > >
> > > > > > > > > [d] Wasserstein, R.L., Schirm, A.L. and Lazar, N.A., 2019. Moving to a world beyond “p< 0.05”.

---

> > > > > > > > > > ### Comment · Reviewer_VWHL · 2021-09-15
> > > > > > > > > > **response to 'We hope the following response solves the concerns.'**
> > > > > > > > > >
> > > > > > > > > > I thank to authors for a passionate discussion  and I appreciate their hard work.
> > > > > > > > > >
> > > > > > > > > > I had a quite detailed discussion with the authors (can be seen above). I believe our discussions were broad but also fruitful. I acknowledge that we have small number of disagreements (I consider these disagreements as intellectual differences and I believe they are a must in science), but our agreements are much more. Hence, I increase my score to 7 and I support the acceptance of the method with extended results and analysis.

---

> > > > > > > > > > > ### Author Response · Authors · 2021-09-16
> > > > > > > > > > > **Thank you for your review**
> > > > > > > > > > >
> > > > > > > > > > > We deeply appreciate the efforts of **R#VWHL**, thank you!

---

### Official Review · Reviewer_XKuz · 2021-07-20

**Rating:** 6
**Confidence:** 5

**Summary:**

For image clustering, this paper extends the mainstream contrastive learning paradigm to a cluster-level scheme, and proposes twin-contrast clustering. Heuristic cluster augmentation equivalents are presented to enable cluster-level contrastive learning. Besides, this paper also derives the evidence lower-bound of the instance-level contrastive objective with the assignments. Experiments on several datasets demonstrate its superiority.

**Limitations And Societal Impact:**

Yes

**Main Review:**

Pros:

(1) The overall writing and presentation are good.

(2) The idea for cluster-level augmentation and contrastive learning is interesting.

(3) Experimental results are good.

Cons:

(1) How to achieve the irrelevant minorities augmentation? Where does the small proportion of irrelevant data come from? Please give more explanation.

(2) The backbone might be different from other baselines. Please analyze the difference and its influence.

(3) Please analyze the influence of training epochs. I notice that at least 1,000 epochs are needed for training, which is much larger than other methods.

(4) Compared with the improvement on CIFAR and ImageNet-Dog, the results on STL-10 and ImageNet-10 are not satisfying. The explanation in subsection 5.3 is so simple. Please give more analysis about this.

(5) In experiments, the largest dataset only contains 20 clusters. Will the proposed method be effective on datasets with more categories? More experiments are suggested.


**Time Spent Reviewing:**

4

---

> ### Author Response · Authors · 2021-08-10
> **We thank the reviewer for the constructive suggestions**
>
> We thank **R#XKuz** for the constructive suggestions. Our response is as follows.
>
> # Q1 How to Achieve the Irrelevant Minorities Augmentation?
> Sorry for the vague statements. As discussed in L115, we actually find an alternative to inject a small proportion of irrelevant data to the cluster representation with Eq. (4). TCC performs soft aggregation on each cluster, i.e., $\sum_i\pi_i(k)f(x_i)$.
>
> * As the softmax product $\pi_i$ is always positive, those data that are not very related to the given cluster still contribute to the cluster's representation, which compiles the **irrelevant**.
>
> * Meanwhile, these irrelevant data are **not** dominating the value of the cluster representation, because the small value of $\pi_i(k)$ scales the feature magnitude during aggregation, which counts the **minority**.
>
> Thus, we can link this augmentation with a very intuitive and simple implementation in the context of set representations. The corresponding statements in the manuscript will be revised in the future version.
>
> # Q2 Backbones and CNNs
> Actually, we adopt the very standardized choice of ResNet-34 [25] and the weakest resolution in training for a fair comparison. As discussed in Sec. 5.2 and Supp. Sec. C, our backbone structure is identical to the most recent clustering models [52, 70].
>
> Furthermore, TCC is evaluated under the strictest setting for a fair comparison against all cited baselines. As depicted in Tab. 1, we follow MiCE [70] and many other articles [9, 75] to cap the input resolution to $96\times96$, while some others use a higher resolution of $224\times224$ on STL and ImageNet. As such, the performance gain can be maximally credited to the design of our model rather than the capacity of the network.
>
> Since the choice of the CNN architecture is mostly unified in the recent papers while the input resolutions vary from different works, we believe analyzing the influence of different input resolutions would be of more value regarding the backbone. The results are shown as follows.
>
> |STL-10 (in \%)|NMI|ACC|ARI|ImageNet-10 (in \%)|NMI|ACC|ARI|
> |:--|:-:|:-:|:-:|:--|:-:|:-:|:-:|
> |MiCE [70] (Low Res.)|63.5|75.2|57.5||-|-|-|
> |CC [52] (High Res.)|76.4|85.0|72.6||76.4|85.0|72.6|
> |TCC (Low Res.)|73.2|81.4|68.9||73.2|81.4|68.9|
> |**TCC (High Res.)**|79.5|87.1|73.9||87.3|92.7|87.0|
>
> **Note that only the results on the last line of the table are new to our manuscript.** Our performance is significantly boosted on ImageNet-10 when having a resolution of $224\times 224$. This is not an abnormal phenomenon. It takes the full resolution on ImageNet to convey all detailed semantics and the subset categories are well-selected. **However, these results do not yield a fair comparison against [9, 70, 75].**
>
> To make our manuscript both concise and illustrative, we will add the comparisons in terms of high-resolution settings to the supplementary material.
>
> # Q3 Training Epochs
> Actually, TCC does not require more epochs or time for training. We set the max epoch to 1000 only as of the conventional stopping criterion, in alignment with MiCE [70] and CC [52]. The details are given in Sec. 5.5.
> * According to Fig. 5 (a), TCC only requires about 250-350 epochs to obtain the best-performing results, while MiCE needs to have at least 600 epochs.
> * As shown in Fig. 5 (b), training 1000 epochs with TCC takes 15 hours on a single V100 GPU, while MiCE needs 21 hours on CIFAR-10. As discussed in L338-345, this is mainly caused by the difference in the complexities when computing the losses.
> * In addition, CC requires around 70 GPU-hours to run 1000 epochs. Though the device might be different to ours, we believe its training time will not be significantly less than ours, providing the same computational resource.
>
> # Q4 STL-10 and ImageNet-10
> As stated in the caption of Tab. 2, CC uses a much higher resolution of $224\times 224$ than our TCC during training, leading to exceptionally high performance.
>
> It is important to mention that the conventional resolution for deep clustering evaluation on these two datasets is $96\times96$ [9, 70, 75], and we opt to follow this practice for a fair comparison. We show the results when TCC is trained with $224\times 224$ images in **Q2**, where TCC becomes the best-performing model on STL-10 and ImageNet-10. This largely endorses our speculation regarding the performance difference. We will clarify this in Sec. 5.3 and point readers to the supplementary material for more detailed results after revision.
>
> Although the original image size on STL-10 is $96\times 96$, resizing it to $224\times 224$ still results in better clustering performance. This is because the last $2\times 2$ pooling layer on a larger feature map preserves more semantic information.
>
> It is interesting that even with a low resolution, TCC still performs well on ImageNet-Dog. This suggests that TCC might have a higher performance lower bound than the compared methods. Evaluating and measuring this ability will be one of our future work.
>
> # Q5 More Categories
> We strongly agree that results with more categories are valuable. We will add a set of comparisons on Tiny-ImageNet in the main body and the results are shown in **Q1** of **R#sHMT**.
>
> These results were not initially reported in the manuscript because
> * A lot of baselines did not include this experiment, and we would like to keep our content concise.
> * In practice, tens of categories already cover a large proportion of real applications.

---

### Official Review · Reviewer_sHMT · 2021-07-25

**Rating:** 6
**Confidence:** 4

**Summary:**

This paper proposed a new clustering method called Twin-Contrast Clustering(TCC), which extends the mainstream contrastive learning paradigm to a cluster-level scheme. TCC simultaneously learns instance- and cluster-level representations by leveraging cluster assignment variables. And TCC uses Gumbel Softmax to reparametrize the assignment variables, which makes TCC is able to be trained end-to-end without auxiliary steps.

**Limitations And Societal Impact:**

Please refer to Cons in Main Review.

**Main Review:**

Pros:
1.	This paper is well-written and easy to follow. Most of the details are fully illustrated and the ablation study is quite adequate.
2.	The network structure is very cleverly designed. The authors give in-depth thoughts on how to calculate the clustering level representation and how to introduce clustering information into the instance level contrastive learning. Gumbel Softmax is also a reasonable approach to obtain a more discrete category representation.

Cons:
1.	Is TCC more limited on large-scale data with more categories? According to the description in the methods section, batch_size=32xK and queue_size L=100xK, so the complexity of the algorithm and the memory requirements increase significantly when there are more categories. It would be helpful to experiment with data such as TinyImageNet with 200 classes in CC or ImageNet-100 for the main conclusion of the paper.
2.	From the experimental results, cluster-level augmentation has a large impact on the final performance, but the explanation of cluster-level aug(b) is not sufficient. From the descriptions of Line113-115 and Line299-301, aug(b) plays the role of using the default soft assignment instead of hard assignment, so how does hard assignment work? Is only the maximum probability feature retained?
3.	The paper lacks a comparison of TCC with the best unsupervised methods, especially DeepCluster-V2 and SwAV, which introduce the concept of clustering. From aug(b) and the larger $\lambda$ used in Gumbel Softmax, TCC still needs a soft semantic representation to help learning, which plays a similar effect to the prototype used in SwAV, so SwAV is a worthwhile baseline for comparison.
4.	Finally a small question, given that most methods deal with balanced category data, and TCC introduces a 1/K posterior probability prior, does this mean that TCC does not perform as robust as methods like CC when dealing with category unbalanced data?


**Time Spent Reviewing:**

5

---

> ### Author Response · Authors · 2021-08-10
> **We appreciate the constructive suggestions**
>
> We appreciate the constructive suggestions provided by **R#sHMT**. Our response is as follows.
>
> # Q1 More Categories
> In practice, we find that many real-world applications only involve a limited number of clusters. Thus, under such a small number of clusters, we set $L=100K$ and batch\_size$=32K$ to ensure a reasonable size of the cluster-level memory bank and batch size. As shown in Fig. 4 (d), even when halving the batch size, TCC still delivers the best performance on CIFAR-10. The size of the memory bank is not a big problem as it is not involved in BP. However, please note that our TCC is not limited by the number of clusters. For example, even a 100-way clustering problem still requires a smaller memory bank size than that of MoCo [24], and the corresponding batch size (3200) can still be handled by the devices nowadays.
>
> **Results with Large Cluster Numbers.** We strongly agree that the experimental results on Tiny-ImageNet are valuable. We haven't reported them in the manuscript because a lot of baselines did not include this experiment. Please find below a glimpse of our results, which will be included in the updated version.
>
> |Tiny-ImageNet (in \%)|NMI|ACC|ARI|
> |---|---|---|---|
> |DCCM [75]|22.4|10.8|3.8|
> |CC [52]| 34.0|14.0|7.1|
> |TCC|43.5|30.6|15.2|
>
> **On the Complexity of TCC.** As discussed in L174-178, the time complexity of TCC's loss is $O(L+J)$. Considering the fact that $L$ is usually much smaller than $J$, our complexity is much lower than one of the state-of-the-art memory-bank approaches, i.e., MiCE [70] with a complexity of $O(KJ)$.
>
> # Q2 Describing the Ablation Baseline (vi) on Augmentation (b)
> We apologize for the vague descriptions and clarify them here. For the hard assignment in Baseline (vi) of Tab. 3, we firstly compute $\text{argmax}_k q(k|x)$ for each image $x$, and then assign it to the corresponding cluster. A cluster representation $r$ is then obtained by directly summing up the image features assigned to it without weighting them. In this way, the irrelevant data according to the model prediction have no contribution to a given cluster. Literally, we call this baseline *without Irrelevant Minorities*. Note that this baseline does not influence the instance-level objective $L_2$, since we are studying the behaviour of cluster-level augmentation.
>
> This baseline then disables backpropagation through $d L_1/d\pi$ as the argmax operator is involved. However, the model is yet trainable, since $dL_1/d f_\theta$ remains and the inference model $q(k|x)$ can still be optimized through $dL_2/d\pi$ with the help of the reparametrization trick.
>
> Based on your suggestion, we will add a standalone section in the supplementary material, providing more training details of all ablation baselines.
>
> # Q3 DeepCluster-V2 and SwAV [7]
>
> We totally agree that comparing TCC with SwAV [7] can be interesting, but this was intentionally omitted from our original submission. The main reason is discussed in L211-213, both DeepCluster-V2/SwAV [7] and [51] are pre-text models and proposed for a different application rather than clustering. They literally involve a clustering stage as a pre-training module to facilitate downstream tasks such as detection and segmentation, while we simply focus on a model dedicated to clustering. We also notice it is a common practice for the latest clustering articles [28,29,52,60] not to compare pre-text baselines like SwAV [7].
>
> On the other hand, we have included one contrastive pre-text baseline with MoCo [24] and k-means in Tab. 2, where we can see that TCC clearly outperforms the baseline. This can be a more related baseline as TCC also employs memory banks. We will revise the paper and add the results of DeepCluster-V2 and SwAV [7] to the main body. The initial comparisons are shown below.
>
> |CIFAR-10 (in \%)|NMI|ACC|ARI|
> |---|---|---|---|
> |DeepCluster-V2|65.2|72.7|58.9|
> |SwAV|71.1|79.6|64.9|
> |TCC|79.0|90.6|73.3|
>
> We re-implement these baselines with the same CNN backbone as TCC, and train them at the same resolution as we described in Tab. 1. It can be observed that these baselines obtain on-par performance against some recent methods. Though conceptually including clusters in training, they still mainly focus on better instance-level general representation learning, instead of improving the clustering assignments or preserving more cluster-level semantics.
>
> # Q4 Imbalanced Categories?
>
> Assuming an even assignment distribution throughout the dataset seems to be a common practice in the recent works including CC [52] (with an entropy-like regularization) and MiCE [70] (with a fixed prior network). The main motivation behind this is to avoid the missing assignments to a certain cluster, i.e., the *degeneracy issue* [70], instead of favouring the current experimental settings.
>
> TCC is able to accept an imbalanced training set. Derived from the instance-level ELBO, our KLD term in Eq. (6) with $p_0=1/K$ just acts as a regularizer rather than a constraint. We are not expecting an exactly uniform posterior produced by the model. Besides, the cluster-level representation aggregated with soft assignment weights is self-normalized by L2-normalization (see Eq. (4)), which means the amplitude of each cluster representation will not shift much even if only a limited number of data are assigned to it in each batch. In other words, an imbalanced set will not influence the NCE computation of Eq. (5).
>
> However, including CC [42], the conventional evaluation protocols do not assess the imbalanced categories issue. So we alternatively show some implicit evidence to endorse the ability of TCC on imbalanced datasets.
> * When removing the KLD term in the loss, TCC still works well, obtaining an **NMI|ACC|ARI** of 76.6|88.9|71.0 on CIFAR-10. We will add this ablation baseline in the future version.
> * It is shown in Fig. 6 that the histogram of the assignments of TCC is not strictly flat. It also depends on the statistics of each batch.

---

> > ### Comment · Reviewer_sHMT · 2021-09-02
> > **re: We appreciate the constructive suggestions**
> >
> > Thanks for the meticulous reply. Most of my concerns have been addressed. In particular, the performance of TCC on Tiny-ImageNet is amazing. Although I still have concerns about the performance of TCC in the face of large models with multi-category data(since the batch size of 32x100=3200 is not a small number), I'll keep my score at 6.

---

> > > ### Author Response · Authors · 2021-09-10
> > > **Thank you for your review**
> > >
> > > We deeply appreciate the efforts and endorsement from **R#sHMT**.
> > >
> > > Let us finally point out that in a many-category case (say 100 categories), a batch size of $32\times100$ is still acceptable for modern GPU devices. When keeping a low resolution, our model even does **not** require distributed training. It is also notable that some contrastive learning models would require larger batch sizes at a scale of many thousands of images [11]. Compared with them, the proposed model does not lead to an unusual device requirement even in the most extreme case. Finally, as stated and experimented in Sec. 5.4, the proposed model still does well when we halve the batch size. The default batch size of $32K$ is just a preferred setting, instead of a hard cap.
> > >
> > > We hope this solves the last concern of **R#sHMT**, and are still willing to answer additional questions.

---

### Decision · Program_Chairs · 2021-09-27

**Decision:**

Accept (Poster)

**Comment:**

The paper proposes a novel extension to contrastive learning, using both cluster level and instance level tasks in representation learning. The bulk of the reviewers found this method interesting and well-motivated, and from my perspective as well using this additional structure is a good direction in contrastive learning / representation learning. The authors felt their work should be viewed independently from SwAV, but the reviewers clearly saw the strong overlap between what the two models do. The reviewers added SwAV as a baseline, and their model performed favorably on their benchmark tasks. However, I encourage the authors to be generous on the connection between their work and SwAV in particular in the final draft, while the downstream applications are similar, I believe the motivations are very similar. Finally, the reviewers requested more benchmarks, such as TinyImagenet, and this satisfied the reviewers. Therefore I recommend acceptance as a poster.

A note that I do believe that [a] "Clustering-friendly Representation Learning via Instance Discrimination and Feature Decorrelation" should be included in the final draft. I do not believe that the significance of the submitted work should be judge by [a], as it is quite new and could have feasibly fallen under the radar with no ill intent. But I believe both papers are doing similar things and as such this paper could do the community a service by including results in [a] and providing this as additional context to their results. I strongly encourage the authors to do this.